# REASONINGBANK: SCALING AGENT SELF-EVOLVING WITH REASONING MEMORY

**Siru Ouyang**[1*], **Jun Yan**[2†], **I-Hung Hsu**[2], **Yanfei Chen**[2], **Ke Jiang**[2], **Zifeng Wang**[2], **Rujun Han**[2],
**Long T. Le**[2], **Samira Daruki**[2], **Xiangru Tang**[3], **Vishy Tirumalashetty**[2], **George Lee**[2],
**Mahsan Rofouei**[4], **Hangfei Lin**[4], **Jiawei Han**[1], **Chen-Yu Lee**[2†], **Tomas Pfister**[2]

[1]UIUC  [2]Google Cloud AI Research  [3]Yale University  [4]Google Cloud AI

siruo2@illinois.edu, {junyann, chenyulee}@google.com

## ABSTRACT

With the growing adoption of large language model agents in persistent real-world roles, they naturally encounter continuous streams of tasks. A key limitation, however, is their failure to learn from the accumulated interaction history, forcing them to discard valuable insights and repeat past errors. We propose REASONINGBANK, a novel memory framework that distills generalizable reasoning strategies from an agent's self-judged successful and failed experiences. At test time, an agent retrieves relevant memories from REASONINGBANK to inform its interaction and then integrates new learnings back, enabling it to become more capable over time. Building on this powerful experience learner, we further introduce memory-aware test-time scaling (MATTS), which accelerates and diversifies this learning process by scaling up the agent's interaction experience. By allocating more compute to each task, the agent generates abundant, diverse experiences that provide rich contrastive signals for synthesizing higher-quality memory. The better memory in turn guides more effective scaling, establishing a powerful synergy between memory and test-time scaling. Across web browsing and software engineering benchmarks, REASONINGBANK consistently outperforms existing memory mechanisms that store raw trajectories or only successful task routines, improving both effectiveness and efficiency; MATTS further amplifies these gains. These findings establish *memory-driven experience scaling* as a new scaling dimension, enabling agents to self-evolve with emergent behaviors naturally arise. Our code can be found at https://github.com/google-research/reasoning-bank.

## 1 INTRODUCTION

The rapid advancement of large language models (LLMs) has significantly accelerated the development of interactive LLM agents (Wang et al., 2024; Liu et al., 2025a), which are crucial in tackling complex real-world tasks that require multi-turn interactions with environments. These agents have demonstrated great potential across diverse scenarios, including web browsing (Gur et al., 2024), computer use (Yang et al., 2024; Xie et al., 2024), and scientific discovery (Ghafarollahi & Buehler, 2025). As these agents are increasingly deployed in persistent, long-running roles, they naturally encounter a continuous stream of tasks and interactions. However, a critical limitation persists: they largely fail to learn from this accumulated experience. By approaching each new task in isolation, they are doomed to (i) repeat similar errors observed in the past (Yin et al., 2025), (ii) discard valuable insights gained from related problems, and, most importantly, (iii) lack self-evolving capabilities that make the agent system more capable over time (Gao et al., 2026). This phenomenon highlights the necessity of building memory-aware agent systems that could learn from their past experiences (Zhang et al., 2025b).

---

[*]This work was done while Siru interned at Google Cloud AI Research.

[†]Corresponding authors.

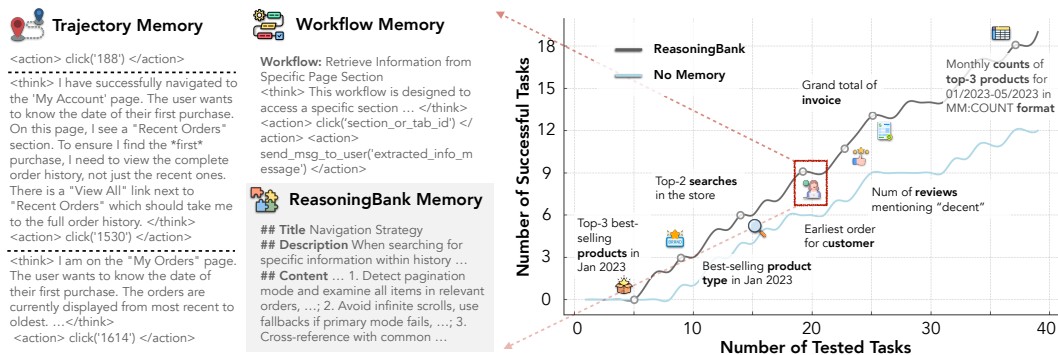

Figure 1: REASONINGBANK induces reusable reasoning strategies, making memory items more transferrable for future use. This enables agents to continuously evolve and achieve higher accumulative success rates than the "No Memory" baseline on the WebArena-Admin subset.

Recent efforts on agent memory (Huang et al., 2026) have primarily focused on storing past interactions for reuse (Zhao et al., 2024; Tang et al., 2025b; Chen et al., 2025; Sun et al., 2025). However, these approaches are often limited to leveraging raw trajectories (Zheng et al., 2024; Kagaya et al., 2024; Kong et al., 2025) or successful routines (i.e., workflows, procedures) (Wang et al., 2025d; Fang et al., 2025). They suffer from two fundamental drawbacks. First, they lack the ability to distill higher-level, transferable reasoning patterns. Second, by over-emphasizing successful experiences, they leave the valuable lessons from an agent's own failures largely underexplored (Zhang et al., 2024). Consequently, existing memory designs often remain limited to passive record-keeping rather than providing actionable, generalizable guidance for future decisions.

To bridge this gap, we propose **REASONINGBANK**, a novel memory framework for agent systems. REASONINGBANK distills and organizes memory items from *both successful and failed experiences* judged by the agent itself without ground-truth labels. As shown in Figure 1, it captures not only effective strategies from successes but also crucial preventative lessons from failures, abstracting them into a collection of actionable principles. This process operates in a closed loop: when facing a new task, the agent retrieves relevant memories from REASONINGBANK to guide its actions. Afterward, the new experience is analyzed, distilled, and consolidated back into the REASONINGBANK, allowing the agent to continuously evolve and improve its strategic capabilities.

With REASONINGBANK as a strong experience learner, we study experience scaling to establish a powerful **synergy between memory and test-time scaling**. Instead of scaling experience through breadth by adding more tasks, we focus on scaling experience through depth by tackling each single task with more exploration. We introduce memory-aware test-time scaling (**MATTS**) in both parallel and sequential settings, which generates diverse exploration to provide contrastive signals, enabling REASONINGBANK to synthesize better memories. It creates a synergy between memory and test-time scaling: high-quality memory steers the scaled exploration toward more promising paths, while the rich experiences generated forge even stronger memories. This positive feedback loop positions **memory-driven experience scaling** as a new scaling dimension for agents.

We conduct extensive experiments on challenging benchmarks for web browsing (WebArena, Mind2Web) and software engineering (SWE-Bench-Verified). We demonstrate that REASONING-BANK outperforms baselines in both effectiveness (up to 20% relative improvement, Table 1) and efficiency (up to 16% fewer interaction steps, Table 1). Additionally, REASONINGBANK synergizes best with MATTS, making it an essential component for memory-driven experience scaling.

Our contributions are threefold: (1) We propose REASONINGBANK, a novel memory framework that distills generalizable reasoning strategies from both successful and failed experiences, beyond prior work that primarily stores raw trajectories or success-only routines. (2) We introduce MATTS that establishes a powerful, bidirectional synergy between memory and test-time scaling, with memory-driven experience as a new scaling dimension. (3) We conduct extensive experiments on web browsing (WebArena, Mind2Web) and software engineering (SWE-Bench-Verified) tasks. We demonstrate that our approaches not only outperform baselines in effectiveness (up to 20% relative improvement) and efficiency (up to 16% fewer interactions), but also uniquely learn from failures and enable agents to develop increasingly complex, emergent reasoning strategies over time.

## 2 RELATED WORK

**Memory for LLM Agents.** Memory has emerged as an essential module in modern agent systems (Zhang et al., 2025b; Hu et al., 2025c; He et al., 2026) to enhance their performance by utilizing past information. Existing memory systems organize and store information in various forms, including plain text (Packer et al., 2023), latent embeddings (Wang et al., 2025b), and structured graphs (Xu et al., 2025; Chhikara et al., 2025; Li et al., 2025b). Beyond memory content, those methods usually involve retrieval mechanisms (e.g., semantic search) with memory management strategies (e.g., updating) (Tan et al., 2025; Hu et al., 2025a). More recently, with the growing development of reinforcement learning (RL) in LLM agents, RL has also been leveraged for memory management in agent systems (Yu et al., 2025a; Zhou et al., 2025). While most efforts primarily emphasize personalization (Zhang et al., 2025a; Zhong et al., 2024) and long-context management (Maharana et al., 2024; Wu et al., 2025; Hu et al., 2025b), this paper falls in the research line of learning from past experiences as memory (Zhao et al., 2024; Su et al., 2025), which is a critical aspect for developing self-evolving agent systems (Gao et al., 2026; Liang et al., 2024). Different from previous works that emphasize reusing successful trajectories (Zheng et al., 2024; Tang et al., 2025a), procedural workflows (Wang et al., 2025d; Qian et al., 2024; Fang et al., 2025; Liu et al., 2025b; Nottingham et al., 2024), or instance-level concepts (Liu et al., 2025c; Suzgun et al., 2025; Zhao et al., 2024), REASONINGBANK stores high-level strategies and reasoning hints. By abstracting experiences into reusable reasoning units, REASONINGBANK enables agents to generalize not only from successful cases (Zhao et al., 2024; Alazraki et al., 2025; Fu et al., 2024) but also by learning from failures, providing richer guidance for test-time learning.

**Agent Test-Time Scaling.** Test-time scaling (TTS) (Snell et al., 2025) has demonstrated strong effectiveness and has become a widely adopted practice in end-to-end problem-solving such as coding (Li et al., 2025a; Yu et al., 2025c) and math reasoning (Muennighoff et al., 2025), where methods including best-of-N (Chow et al., 2025), beam search (Wu et al., 2024b), and leveraging verifiers (Setlur et al., 2025) are commonly employed. However, its application to multi-turn interactive scenarios, particularly agentic tasks, remains underexplored. Existing works mainly adapt the lesson learned from reasoning tasks (Zhu et al., 2025) and scale different dimensions of agentic systems, including the search space for each action (Yu et al., 2025b), the number of agents in multi-agent systems (Jin et al., 2025), and the number of interactions with the environment (Shen et al., 2025). We found that none of these efforts considers the role of *agent memory* in scaling, where an agent can learn from past experiences to guide future decisions. Our work extends this line of research by introducing memory-aware test-time scaling (MATTS). As we will show in our empirical results (§4.3 and §4.4), memory offers benefits beyond mere computational scaling, where memory and scaling synergistically work towards better performance.

## 3 METHODOLOGY

In this section, we introduce the problem setup (§3.1), and present our proposed REASONINGBANK (§3.2), based on which we further develop memory-aware test-time scaling (MATTS) (§3.3).

### 3.1 PROBLEM FORMULATION

**Agent Configuration.** The scope of this work focuses on language model (LM)-based agents. The agent policy $\pi_{\mathcal{L}}(\cdot|\mathcal{M}, \mathcal{A})$ is parameterized by the backbone LLM $\mathcal{L}$, conditioned on a memory module $\mathcal{M}$, and the action space $\mathcal{A}$, denoted as $\pi_{\mathcal{L}}$ for short. The agent needs to perform a task via interacting with the environment, which can be viewed as a sequential decision-making process. Formally, the transition function of the environment is defined as $\mathcal{T}(s_{t+1}|s_t, a_t)$ where $s_t$ is the state and $a_t$ is the action selected by $\pi_{\mathcal{L}}$ at time $t$. We focus on web browsing and software engineering (SWE) tasks. $\mathcal{A}$ is a set of web navigation operations for web browsing and bash commands for SWE, $\mathcal{M}$ is REASONINGBANK and initialized as empty. For each given task, the agent generates a trajectory of $(o_{0:t}, a_{0:t})$ for $t$ steps, where observation $o_t$ is from the current state $s_t$. Observations are text-based accessibility tree of web pages[1] for web browsing tasks and code snippets for SWE. The

---

[1] We use the thinking process of $\pi_{\mathcal{L}}$ as the approximation of $o_{0:t}$ due to lengthy observation representations following Wang et al. (2025d).

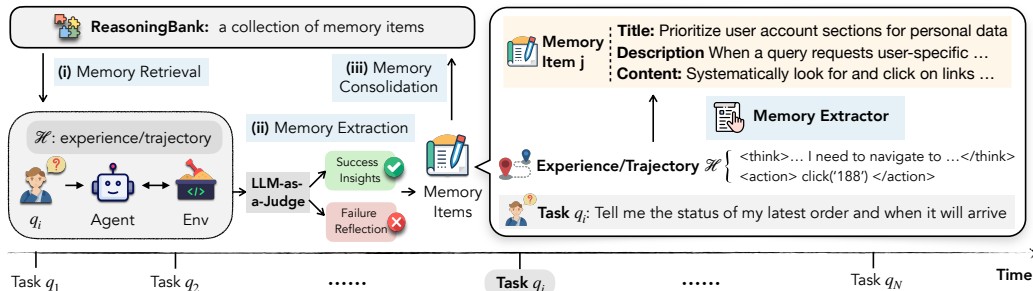

Figure 2: Overview of REASONINGBANK. Experiences are distilled into structured memory items with a title, description, and content. For each new task, the agent retrieves relevant items to interact with the environment, and constructs new ones from both successful and failed trajectories. These items are then consolidated into REASONINGBANK, forming a closed-loop memory process.

agent needs to generate an action $a_{t+1} \in \mathcal{A}$ via $\pi_{\mathcal{L}}(o_{0:t}, a_{0:t}; \mathcal{M}, \mathcal{A}) \rightarrow a_{t+1}$. For implementation, the memory module $\mathcal{M}$ contributes relevant memories as additional system instruction for $\pi_{\mathcal{L}}$.

**Test-Time Learning.** We focus on the test-time learning paradigm (Wu et al., 2024a; Wang et al., 2025c) where a sequence of task queries $\mathcal{Q} = \{q_1, q_2, ..., q_N\}$ arrives in a streaming fashion, i.e., each query is revealed and must be completed sequentially without access to future ones. In this setting, no ground truth is available during test-time, so the agent must continually *evolve* by only leveraging its own past trajectories and any self-verification without relying on external labels. This streaming setting highlights two key challenges: (i) how to extract and preserve useful memory from past trajectories, and (ii) how to effectively leverage such memory for future queries to avoid redundantly re-discovering already successful strategies or repeating past mistakes.

## 3.2 REASONINGBANK

Past raw trajectories (or experiences), while being comprehensive and original, are often too lengthy and noisy to be directly applied to the current user query. As illustrated in Figure 2, REASONINGBANK distills useful strategies and reasoning hints from past experiences into structured memory items, which are then stored in the agent's memory for future reuse.

**Memory Schema.** Memory items in REASONINGBANK are designed and induced from past experiences as structured knowledge units that abstract away low-level execution details while preserving transferrable reasoning patterns and strategies. Each memory item specifies three components: (i) a *title*, which serves as a concise identifier summarizing the core strategy or reasoning pattern; (ii) a *description*, which provides a brief one-sentence summary of the memory item; and (iii) the *content*, which records the distilled reasoning steps, decision rationales, or operational insights extracted from past experiences. Together, memory items extracted are both human-interpretable and machine-usable, facilitating efficient usage and integration with agents.

**Integrating REASONINGBANK with Agents.** An agent $\pi_{\mathcal{L}}$ equipped with REASONINGBANK can draw upon a curated pool of transferable strategies to guide decision-making. This enables the agent to recall effective insights, avoid previously observed pitfalls, and adapt more robustly to unseen queries. The integration proceeds in three steps: (i) *memory retrieval*, (ii) *memory extraction*, and (iii) *memory consolidation*, as shown in Figure 2. During *memory retrieval*, the agent queries REASONINGBANK with the current query context to identify the top-$k$ relevant experiences and their corresponding memory items using embedding-based similarity search. Retrieved items are injected into the agent's system instruction, ensuring that action prediction from $\mathcal{A}$ is grounded with useful past experiences. When the current query task is completed, we will perform *memory extraction* to extract new memory items. The first step is to obtain proxy correctness signals for trajectories: we adopt an LLM-as-a-judge (Gu et al., 2024) to label outcomes as success or failure given the query and trajectory without any ground-truth reference. Based on these signals, we apply different extraction strategies: successful experiences contribute validated strategies, while failed ones supply counterfactual signals and pitfalls that help sharpen guardrails. In practice, we extract multiple memory items for each trajectory/experience as detailed in Appendix A.1. Finally, *memory consolidation* incorporates these items into REASONINGBANK with a simple addition operation, maintaining an evolving repository of memory items. Details are in Appendix A.2. Together, these

Figure 3: Comparison of *(a) vanilla TTS* that independently run REASONINGBANK for multiple times without aggregating on trajectories, MATTS with *(b) parallel scaling*, where self-contrast across multiple trajectories curates reliable memory, and *(c) sequential scaling*, where self-refinement enriches memory with intermediate reasoning signals.

steps form a closed-loop process: the agent leverages past experiences, constructs new memory from current tasks, and continually updates its memory, enabling sustained evolvement in test-time learning scenarios.[2]

### 3.3 MATTS: MEMORY-AWARE TEST-TIME SCALING

REASONINGBANK enables learning from experiences to translate more experiences into greater improvements. As test-time scaling (TTS) (Snell et al., 2025) recently emerged as a powerful strategy for boosting the performance of LLM agents (Zhu et al., 2025), it shows strong potential by allocating additional inference-time computation to generate abundant exploration histories. A vanilla combination is depicted in Figure 3(a), where more trajectories are independently converted to more memory items. However, this is suboptimal because it does not leverage inherent contrastive signal that arises from abundant explorations on the same problem, which limits the resulting performance advantage brought by TTS. To address this, we propose *Memory-aware Test-Time Scaling* (MATTS), a novel integration of test-time scaling with our memory framework, REASONINGBANK. Unlike the vanilla approach, MATTS learns by deliberately aggregating from the abundant successful and failure trajectories generated during scaling for more effective memory curation. We design two complementary instantiations for MATTS, parallel scaling and sequential scaling, as illustrated in Figure 3(b) and 3(c) with detailed implementation in Appendix A.3.

**Parallel Scaling.** In the parallel setting, we generate multiple trajectories for the same query under the guidance of retrieved memory items. By comparing and contrasting (*self-contrast* (Chen et al., 2020)) *across different trajectories*, the agent can identify consistent reasoning patterns while filtering out spurious solutions. This process enables more reliable memory curation from multiple trials of a single query and promotes diverse yet grounded exploration.

**Sequential Scaling.** We iteratively refine its reasoning *within a single trajectory* after the initial completion, following the principle of *self-refinement* (Madaan et al., 2023). In this process, the intermediate notes generated in self-refinement are also used as valuable signals for memory, since they capture reasoning attempts, corrections, and insights that may not appear in the final solution.

We define the scaling factor $k$, denoting the number of trajectories for parallel scaling and refinement steps for sequential scaling. Equipped with REASONINGBANK, both parallel and sequential strategies become memory-aware, ensuring that the additional computation allocated at test time translates into more transferable and higher-quality memory for future tasks.

## 4 EXPERIMENTS

### 4.1 SETUP

Following existing work (Wang et al., 2025d), we conduct experiments on WebArena (Zhou et al., 2024) which features general web navigation across diverse domains[3], and Mind2Web (Deng

---

[2]We deliberately keep the memory usage pipeline simple, avoiding additional complexity in retrieval or consolidation so as to highlight the contribution of REASONINGBANK itself. These components, however, can be further enhanced with more sophisticated techniques, which could provide additional benefits.

[3]We exclude the domain of *Map* due to website issues following Miyai et al. (2025) for a fair comparison.

Table 1: Experiment results of REASONINGBANK and MATTS (parallel scaling, $k = 5$, pass@1) on WebArena benchmark. Success rate (SR ↑) and the number of steps (Step ↓) are reported on 5 subsets for 3 different backbone LLMs.

| Models | Shopping (187) | | Admin (182) | | Gitlab (180) | | Reddit (106) | | Multi (29) | | Overall (684) | |
|---|---|---|---|---|---|---|---|---|---|---|---|---|
| | *SR* | *Step* | *SR* | *Step* | *SR* | *Step* | *SR* | *Step* | *SR* | *Step* | *SR* | *Step* |
| *Gemini-2.5-flash* | | | | | | | | | | | | |
| No Memory | 39.0 | 8.2 | 44.5 | 9.5 | 33.9 | 13.3 | 55.7 | 6.7 | 10.3 | 10.0 | 40.5 | 9.7 |
| Synapse | 40.6 | 7.0 | 45.1 | 9.1 | 35.6 | 13.0 | 59.4 | 6.5 | 10.3 | 10.5 | 42.1 | 9.2 |
| AWM | 44.4 | 7.0 | 46.7 | 8.8 | 37.2 | 13.2 | 62.3 | 6.1 | 3.4 | **7.7** | 44.1 | 9.0 |
| REASONINGBANK | **49.7** | **6.1** | **51.1** | **8.2** | **40.6** | **12.3** | **67.0** | **5.6** | **13.8** | 8.8 | **48.8** | **8.3** |
| +MATTS | 53.0 | 6.3 | 53.8 | 7.6 | 42.8 | 11.9 | 70.8 | 5.4 | 17.2 | 8.0 | 51.8 | 7.9 |
| *Gemini-2.5-pro* | | | | | | | | | | | | |
| No Memory | 45.5 | 7.6 | 51.1 | 8.7 | 35.0 | 11.6 | 71.7 | 6.0 | 6.9 | 8.8 | 46.7 | 8.8 |
| Synapse | 46.5 | 6.6 | 52.2 | 8.9 | 38.3 | 11.3 | 68.9 | 5.9 | 6.9 | 9.0 | 47.7 | 8.5 |
| AWM | 48.1 | 6.4 | 49.3 | 9.8 | 40.0 | 11.2 | 68.9 | 6.4 | 3.4 | 9.3 | 47.6 | 8.7 |
| REASONINGBANK | **51.9** | **6.0** | **56.6** | **7.7** | **44.4** | **9.8** | **80.2** | **5.1** | **13.8** | 8.2 | **53.9** | **7.4** |
| +MATTS | 54.0 | 5.9 | 58.2 | 7.4 | 46.7 | 9.1 | 83.0 | 5.3 | 20.7 | 7.2 | 56.3 | 7.1 |
| *Claude-3.7-sonnet* | | | | | | | | | | | | |
| No Memory | 38.5 | 6.1 | 49.5 | 8.4 | 36.7 | 10.6 | 53.8 | 5.5 | 0.0 | 11.6 | 41.7 | 8.0 |
| Synapse | 39.6 | 5.8 | 50.5 | 8.5 | 38.0 | 10.0 | 53.8 | 6.1 | 0.0 | 11.8 | 42.6 | 7.9 |
| AWM | 39.6 | 7.2 | 47.8 | 9.3 | 34.6 | 10.9 | 52.8 | 7.0 | 0.0 | 12.4 | 40.8 | 8.9 |
| REASONINGBANK | **44.9** | **5.6** | **53.3** | **7.6** | **41.1** | **9.5** | **57.5** | **5.2** | 3.4 | **10.5** | **46.3** | **7.3** |
| +MATTS | 47.1 | 5.8 | 55.5 | 7.4 | 43.3 | 9.4 | 60.4 | 5.0 | 10.3 | 9.1 | 48.8 | 7.2 |

et al., 2023) that tests generalization of agents on versatile operations and environments. We also conduct experiment on SWE-Bench-Verified for repository-level issue-resolving. For comparison, we consider baselines ranging from memory-free agents (No Memory) to trajectory-based memory (Synapse) (Zheng et al., 2024) and workflow-based memory (AWM) (Wang et al., 2025d). Our agents are built on Gemini-2.5 (Comanici et al., 2025) and Claude-3.7 (Anthropic, 2025) models using BrowserGym (de Chezelles et al., 2025) for web browsing and bash-only for SWE, following ReAct (Yao et al., 2023) style with default decoding configurations. We evaluate effectiveness (success rate, SR) and efficiency (average steps, AS), with specific metrics varying for each dataset. Full descriptions for datasets, baselines, implementations, and evaluation are in Appendix B.

## 4.2 RESULTS OF REASONINGBANK

Tables 1, 2, 3 summarize the main evaluation results of REASONINGBANK on WebArena, Mind2Web, and SWE-Bench-Verified accordingly. We have the following observations.

**REASONINGBANK consistently outperforms baselines across LLM backbones on all datasets.** Specifically, REASONINGBANK improves the overall success rate on WebArena (Table 1) by +8.3, +7.2, and +4.6 with three different backbone LLMs compared to memory-free agents. A similar pattern holds on Mind2Web (Table 2), where REASONINGBANK delivers clear gains across cross-task, cross-website, and cross-domain settings, underscoring both the consistency and scalability of its benefits across datasets and model sizes. Results on SWE-Bench-Verified (Table 3) further confirm its robustness. Crucially, unlike baselines such as Synapse and AWM that rely on a narrow, homogeneous memory source derived exclusively from successful trajectories, REASONINGBANK employs a superior extraction strategy that is key to its consistent out-performance.

**REASONINGBANK enhances generalization with better transferrable memory across tasks.** We also evaluate in challenging generalization settings. On WebArena (Table 1), the *Multi* subset requires transferring memory across multiple websites, where REASONINGBANK achieves a notable gain of +4.6 averaged SR over the strongest baseline. In contrast, strong baselines such as AWM fail to provide gains and even degrade in this setting. On Mind2Web (Table 2), which includes cross-task, cross-website, and cross-domain evaluations that impose progressively higher demands, REASONINGBANK consistently improves success rates. The gains are especially pronounced in the cross-domain setting, which requires the highest level of generalization. These results demonstrate

Table 2: Results on Mind2Web benchmark for cross-task, cross-website, and cross-domain generalization test. EA ($\uparrow$) is short for element accuracy, $AF_1$ ($\uparrow$) is short for action $F_1$, and SSR ($\uparrow$) is short for step success rate. SR ($\uparrow$) is the task-level success rate measuring if all steps are correct.

| Models | Cross-Task | | | | Cross-Website | | | | Cross-Domain | | | |
|---|---|---|---|---|---|---|---|---|---|---|---|---|
| | EA | $AF_1$ | SSR | SR | EA | $AF_1$ | SSR | SR | EA | $AF_1$ | SSR | SR |
| *Gemini-2.5-flash* | | | | | | | | | | | | |
| No Memory | 46.0 | 59.1 | 40.3 | 3.3 | 39.8 | 45.1 | 31.7 | 1.7 | 35.8 | 37.9 | 31.9 | 1.0 |
| Synapse | 47.0 | 59.5 | 41.2 | 3.5 | 40.3 | 46.0 | 32.1 | 1.9 | 36.3 | 38.5 | 32.4 | 1.1 |
| AWM | 46.3 | 56.1 | 41.0 | 3.5 | 39.1 | 42.2 | 31.7 | 2.1 | 33.3 | 36.5 | 30.1 | 0.7 |
| REASONINGBANK | **52.1** | **60.4** | **44.9** | **4.8** | **44.3** | **52.6** | **33.9** | **2.3** | **40.6** | **41.3** | **36.6** | **1.6** |
| *Gemini-2.5-pro* | | | | | | | | | | | | |
| No Memory | 49.3 | 60.2 | 44.4 | 3.5 | 41.2 | 49.8 | 34.8 | 3.4 | 37.9 | 37.7 | 35.0 | 1.4 |
| Synapse | 50.1 | 61.0 | 44.7 | 3.6 | 41.8 | 51.2 | 35.0 | 3.2 | 38.5 | 39.8 | 35.6 | 1.5 |
| AWM | 48.6 | 61.2 | 44.4 | 3.7 | 41.9 | 47.9 | 34.8 | 2.3 | 37.3 | 38.1 | 34.4 | 1.2 |
| REASONINGBANK | **53.6** | **62.7** | **45.6** | **5.1** | **46.1** | **54.8** | **36.9** | **3.8** | **42.8** | **45.2** | **38.1** | **1.7** |

that memory curated by REASONINGBANK is more robust and transferable, enabling agents to generalize effectively across diverse scenarios.

**REASONINGBANK achieves superior efficiency by leveraging past experiences as memory.** In addition to higher success rates, REASONINGBANK also reduces the number of interaction steps needed to complete tasks, as shown in the *Step* metric of Table 1 and 3. On WebArena, across almost all subsets and backbones, REASONINGBANK lowers the average step count by up to 1.4 compared with "No Memory", and 1.6 compared with other memory baselines. The average step on SWE-Bench-Verified is also smaller by saving 2.8 and 1.3 steps respectively. This indicates that REASONINGBANK enables agents to solve tasks more efficiently by reusing and refining reasoning knowledge, thus avoiding unnecessary or redundant exploration.

Table 3: Experiment results of REASONINGBANK on SWE-Bench-Verified dataset for issue-resolving in a given repository.

| Methods | Resolve Rate | AS |
|---|---|---|
| *Gemini-2.5-flash* | | |
| No Memory | 34.2 | 30.3 |
| Synapse | 35.4 | 30.7 |
| REASONINGBANK | **38.8** | **27.5** |
| *Gemini-2.5-pro* | | |
| No Memory | 54.0 | 21.1 |
| Synapse | 53.4 | 21.0 |
| REASONINGBANK | **57.4** | **19.8** |

### 4.3 RESULTS OF MATTS

We first present the overall results of MATTS on WebArena in Table 1, which demonstrate additional strong performance improvement and efficiency gains with REASONINGBANK.[4] To study the scaling effect in-depth with both parallel and sequential variants, we experimented MATTS with Gemini-2.5-flash on Webarena-Shopping subset. To investigate the overall scaling effect, we benchmark with **(i) MATTS w/o memory**, which represents the scaling setting without memory mechanism, **(ii) MATTS w/o aggregation**, which is equal to Vanilla TTS in Figure 3(a) and **(iii) MATTS** to demonstrate the effect with respect to scaling factor $k$. Notably, $k = 1$ is the setting without scaling. For parallel scaling, we compute Best-of-N (BoN) as the final metric detailed in Appendix A.3. Results are shown in Figure 4.

**Both parallel scaling and sequential scaling boost performance.** Increasing $k$ generally improves success rate, confirming the benefit of allocating more inference-time computation. With MATTS, parallel scaling grows from 49.7 ($k = 1$) to 55.1 ($k = 5$), while sequential scaling rises from 49.7 to 54.5. For the baseline of MATTS w/o memory, the gains are smaller and less consistent (e.g., parallel scaling fluctuates between 39.0 and 42.2 and sequential scaling fluctuates between 37.4 and 40.6). In contrast, MATTS enables stronger and more stable improvements across both scaling strategies, highlighting its role in making scaling more effective.

**MATTS is consistently better than vanilla TTS.** With REASONINGBANK, MATTS consistently surpasses MATTS w/o aggregation (vanilla TTS), showing that memory-aware coordination and aggregation is important. Specifically, at $k = 5$, MATTS achieves 55.1 in parallel scaling compared

---

[4]By default, MATTS integrates REASONINGBANK, but it could also use other memory mechanisms.

with 52.4 for vanilla TTS, and 54.5 versus 51.9 in sequential scaling. These improvements highlight that memory-aware scaling effectively directs the agent toward more promising solutions by synthesizing insights from multiple trajectories or interaction steps to leverage contrastive signals.

**Sequential scaling shows short-term advantage, but parallel dominates at larger scales for REASONINGBANK.** With stronger memory mechanisms such as REASONING-BANK, sequential refinement brings higher gains at small $k$, but its benefit quickly saturates—once the model either succeeds or fails decisively, further refinements add little new insight. In contrast, parallel scaling continues to provide diverse rollouts that allow the model to critique and improve upon its own generations, leading it to surpass sequential at larger $k$ (e.g., 55.1 vs. 54.5 at $k = 5$). In contrast, for vanilla TTS which is not equipped with memory module, sequential scaling yields little or even no benefit as scaling goes on, and parallel scaling consistently dominates.

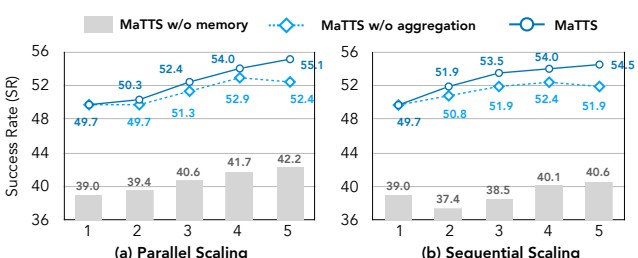

Figure 4: Effect of scaling factor $k$ for MATTS under with REASONINGBANK on WebArena-Shopping subset. We compare (a) parallel and (b) sequential test-time scaling.

### 4.4 SYNERGY OF MEMORY AND TEST-TIME SCALING

While the previous section establishes the overall effectiveness of MATTS, we highlight the synergy between memory and TTS in this section. Figure 5 presents a snapshot of MATTS on the WebArena-Shopping subset with parallel scaling factor $k = 5$, where we report both Pass@1 (randomly selected trajectory) and Best-of-5 (BoN). This setting allows us to examine the bidirectional interaction between memory quality and scaling effectiveness.

**Better memory enables stronger test-time scaling performance.** To see how memory improves the effectiveness of scaling, we focus on the BoN results, which directly measure an agent's ability to surface the best outcome among multiple rollouts. As shown by blue bars in Figure 5, the benefit of scaling depends critically on the underlying memory. Without memory, scaling yields slight improvement, with BoN rising only from 39.0 to 42.2. Weaker memory mechanisms such as Synapse and AWM provide moderate gains, reaching 44.4 and 47.6, respectively. In contrast, MATTS with REASONINGBANK delivers the strongest benefit, with BoN climbing from 49.7 to 55.1. These results show that high-quality memory directs scaling toward more promising rollouts, ensuring that the additional trajectories are not wasted but converted into higher success rates.

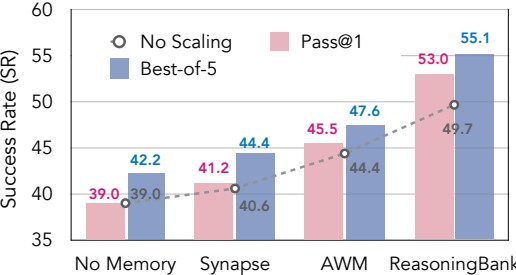

Figure 5: Snapshot of MATTS on WebArena-Shopping subset with different memory mechanisms with $k = 5$. We compute BoN for all 5 trajectories and Pass@1 with one randomly selected trajectory.

**Scaling yields better memory curation.** To fairly evaluate how scaling feeds back into memory, we report Pass@1, which measures the average quality of trajectories after memory curation and allows direct comparison with the no-scaling case. The trend is depicted in pick bars and is striking: scaling actually reduces performance for weaker memories, where Synapse slightly increases from 40.6 to 41.2, and AWM from 44.4 to 45.5. These phenomenon suggest that without strong guidance, the extra rollouts generated by scaling offer diminishing returns. In contrast, REASONINGBANK provides far more benefits: Pass@1 rises from 49.7 to 53.0, showing that high-quality memory can harness the diversity of scaling to extract constructive contrastive signals. This asymmetry highlights that scaling alone is insufficient; only when paired with good memory mechanism does it contribute to curation of more effective memory, closing the virtuous cycle.

## 5 ANALYSIS

We analyze REASONINGBANK through several aspects: incorporating failure trajectories, examining emergent strategies, and evaluating efficiency across both successful and failed cases. Additional analyses are presented in Appendix C, including but not limited to number of retrieved experiences, additional results on smaller open-source model, and inference cost study.

**Emergent behaviors with REASONINGBANK.** We find that the strategies in REASONINGBANK are not flat or monolithic, but instead evolve over time, exhibiting emergent behaviors that resemble the learning dynamics of RL (Wang et al., 2025a). As illustrated in a human case study in Figure 8, memory items describing a specific strategy "*User-Specific Information Navigation*" in REASONINGBANK could gradually evolve during test-time learning process. It starts from execution-oriented or procedural strategies (e.g., find navigation links), where the agent follows straightforward action rules. It then progresses to adaptive self-reflections such as re-verifying identifiers to reduce simple mistakes. With more experiences, the same memory item evolves into adaptive checks, where the agent systematically leverages available search or filters to ensure completeness before results. Finally, it eventually matures into compositional strategies such as cross-referencing task requirements and reassessing options. This evolution highlights how REASONINGBANK enables agents to refine strategies from low-level actions to high-level reasoning during test-time learning.

**REASONINGBANK makes good use of failure trajectories.** Figure 6 compares different memory designs on WebArena-Shopping with Gemini-2.5-flash under two settings: using only successful trajectories versus leveraging both successes and failures. Baseline methods such as Synapse and AWM build memory solely from successful trajectories, and thus are not equipped to benefit from failures. As a result, when failures are added, their performance is limited or even degraded: Synapse increases only from 40.6 (success only) to 41.7 (with failures), while AWM drops from 44.4 to 42.2. In contrast, the design of REASONINGBANK enables distillation of reasoning patterns from *both* successes and failures, achieving 46.5

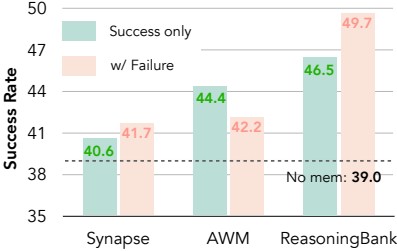

Figure 6: Ablation results of incorporating failure trajectories for memory induction.

on success-only traces and further improving to 49.7 when failures are included. This highlights that, unlike baselines, REASONINGBANK can transform failures into constructive signals rather than noise, enabling more robust generalization.

**REASONINGBANK exhibits robustness to LLM-as-a-Judge calibration.** A critical step in our method is sourcing proxy correctness signals for agent trajectories via an LLM-as-a-Judge. Here we quantitatively calibrate this judge and analyze its robustness to verification noise. We conduct our analysis on the WebArena-Shopping subset, using Gemini-2.5-flash as the judge. We first establish the baseline accuracy by comparing the judge's predictions against ground-truth labels, which we find to be 72.7%. To systematically study the robustness of REASONINGBANK to the judge quality, we simulate different levels of verification accuracy. This is achieved by probabilistically correlating the judge's labels with the ground-truth. For example, a 100% accurate verifier uses the ground-truth

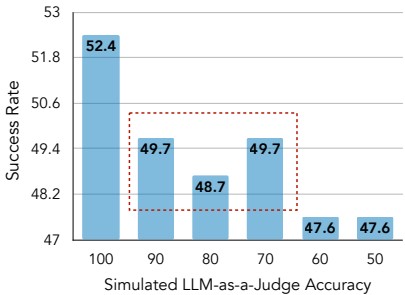

Figure 7: Results for SR w.r.t. simulated LLM-as-a-judge accuracy.

labels directly. A 90% accurate verifier is simulated by using the correct (ground-truth) label 90% of the time and the incorrect (flipped) label 10% of the time. We extend this simulation down to 50% accuracy, which represents a random-guess baseline for this binary (success/failure) classification task. The results are presented in Figure 7. We observe that the judge's accuracy does not significantly impact the performance of REASONINGBANK, as all variants achieve similar success rates within reasonable accuracy range (70%-90%). Intuitively, the 100% (ground-truth)

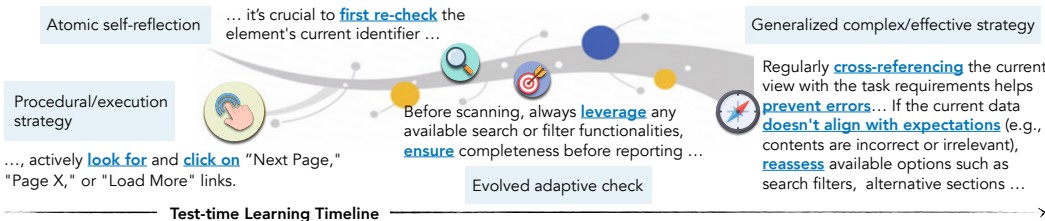

Figure 8: A case study illustrating emergent behaviors in REASONINGBANK through memory items.

Table 4: Average number of steps on successful and failed test instances across four WebArena domains. REASONINGBANK consistently reduces the number of steps compared to the vanilla baseline, with notably larger reductions on successful instances.

| Models | Shopping | | Admin | | Gitlab | | Reddit | |
|---|---|---|---|---|---|---|---|---|
| | *Successful* | *Failed* | *Successful* | *Failed* | *Successful* | *Failed* | *Successful* | *Failed* |
| No Memory | 6.8 | 8.7 | 8.4 | 10.4 | 8.6 | 15.7 | 6.1 | 7.6 |
| REASONINGBANK | 4.7↓2.1 | 7.3↓1.4 | 7.0↓1.4 | 9.5↓0.9 | 7.6↓1.0 | 15.5↓0.2 | 5.0↓1.1 | 6.8↓0.8 |

accuracy setting yields the best performance. These findings confirm that REASONINGBANK is robust to noise in the verification step.

**REASONINGBANK delivers targeted efficient gains.** While the overall number of steps in Table 1 provides a general view of model efficiency, it does not distinguish whether reductions come from successful or failed trajectories. To gain deeper insight, we further separate the analysis into successful and failed test cases, which allows us to understand the source of step reduction: a desirable system should reduce unnecessary exploration when it is on the right track, rather than merely cutting short failed attempts. The results are shown in Table 4. We find that REASONINGBANK consistently reduces the number of steps across all domains compared to the baseline. More importantly, the reduction is particularly pronounced on successful cases, reaching up to 2.1 fewer steps (a 26.9% relative reduction) than on failed ones. This indicates that REASONINGBANK primarily helps the agent reach solutions with fewer interactions by strengthening its ability to follow effective reasoning paths rather than simply truncating failed trajectories, which highlight the role of memory in guiding purposeful decision-making and improving efficiency in practice.

## 6 CONCLUSION

We introduce REASONINGBANK, a memory framework that distills strategy-level reasoning signals from both successes and failures and integrates them into test-time scaling (MATTS). Extensive experiments show that REASONINGBANK consistently improves performance while reducing redundant exploration. Further results reveal a strong synergy between memory and scaling: REASONINGBANK guides scaling toward more promising rollouts, while diverse rollouts enrich memory with valuable contrastive signals. We also provide analyses of individual components and emergent behaviors. Our findings suggest a practical pathway toward building adaptive and lifelong-learning agents, with additional future directions and limitations in Appendix D and E.

## ACKNOWLEDGEMENT

We thank Jiao Sun, Jing Nathan Yan, Chi Wang, and members from Google Cloud AI Research for their valuable feedback during the preparation of the paper. Siru was supported by the Molecule Maker Lab Institute: An AI Research Institutes program supported by NSF under Award No. 2019897.

ETHICS STATEMENT

This work does not involve human subjects, sensitive personal data, or any tasks requiring Institutional Review Board (IRB) approval. All datasets used in our experiments (WebArena, Mind2Web, SWE-Bench-Verified) are publicly available and widely used in the community; detailed descriptions and preprocessing steps are provided in Appendix B to ensure transparency and compliance. Our methods, REASONINGBANK and MATTS, focus on improving the effectiveness, efficiency and generalization of LLM-based agents in web navigation and software engineering tasks. While these advances aim to benefit research on adaptive and lifelong-learning agents, we are mindful of potential concerns regarding misuse (e.g., automation of harmful tasks). We therefore restrict our experiments to established academic benchmarks and commit to open-sourcing our code under a research-only license to encourage responsible use. We have carefully followed the ICLR Code of Ethics throughout this research and submission process.

REPRODUCIBILITY STATEMENT

We have made extensive efforts to ensure the reproducibility of our work. The full methodology of REASONINGBANK and Memory-Aware Test-Time Scaling (MATTS) is detailed in Section 3, with additional implementation details provided in Appendix A. Experimental setups, including datasets (WebArena, Mind2Web, SWE-Bench-Verified), baselines, evaluation protocols, and environment configurations, are described in Section 4.1 and Appendix B. For clarity, we provide ablation studies (Section 5), efficiency analyses (Section 5), and emergent behavior case studies (Section 5) to validate the robustness of our findings. To further facilitate replication, we include precise descriptions of the memory schema, retrieval, construction, and consolidation pipeline in Section 3.2 and Appendix A.1-A.2, and we outline the scaling procedures for both parallel and sequential variants in Section 3.3 and Appendix A.3. We have released our full codebase to the open-source community to foster transparency, reproducibility, and future research.

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

**Contents of Appendix**

# A EXPERIMENT DETAILS

This section details the implementation of REASONINGBANK with agent systems mentioned in Section 4.1 for web browsing tasks including WebArena and Mind2Web. We first present all the prompts used for memory extraction in Appendix A.1, and then we provide the technical details for memory extraction, retrieval, and consolidation in Appendix A.2.

## A.1 PROMPTS USED FOR REASONINGBANK

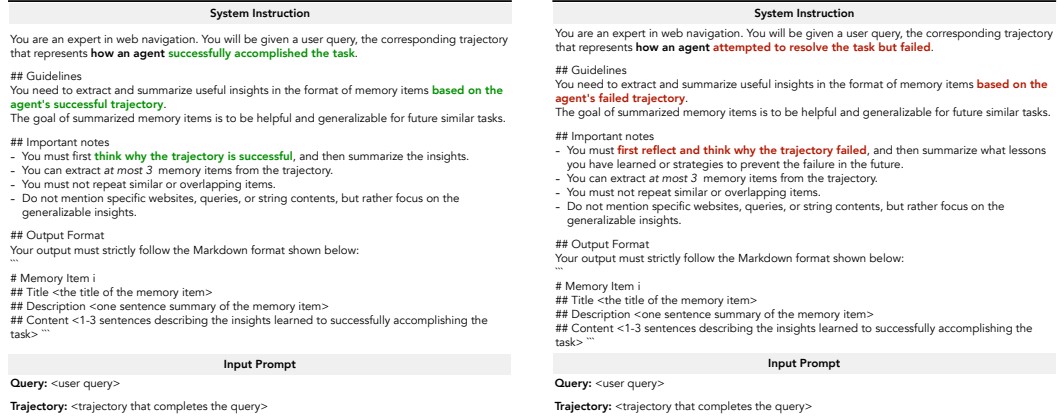

Figure 9: System instructions for extracting memory items from agent trajectories: the left panel targets successful trajectories (summarizing why they succeed), while the right targets failed trajectories (reflecting on failure and deriving lessons).

**Memory Extraction.** Figure 9 illustrates the system instructions we used to guide the extraction of memory items from agent trajectories mentioned in Section 3.2. We will first obtain correctness signals from LLM-as-a-Judge (Gu et al., 2024) using the same backbone LLMs. When the trajectory corresponds to a successful case (left panel), the instruction emphasizes analyzing why the trajectory led to success and summarizing transferable reasoning strategies. Conversely, when the trajectory represents a failed case (right panel), the instruction requires reflecting on the causes of failure and articulating lessons or preventive strategies. In both settings, the output format is constrained to at most three memory items expressed in a structured Markdown format, ensuring that the resulting insights are concise, non-redundant, and generalizable across tasks rather than tied to specific websites or queries.

**LLM-as-a-Judge for Correctness Signals.** Figure 10 displays the instruction used for self-evaluation used to get binary signals for both successes and failures. Given the current user query, trajectory in resolving the query, final state of the website, and model output, the LLM is required to output the state of "Success" or "Failure" of whether the trajectory given successfully resolved the query or not.

## A.2 IMPLEMENTATION DETAILS

**Memory Extraction.** We use an LLM-based extraction pipeline to convert raw trajectories into structured memory items. Specifically, we design a prompt template that asks the model to distill reasoning patterns into three components: *title*, *description*, and *content* as previously mentioned in Appendix A.1. The backbone LLM of the extractor is set to the same as the agent system with temperature 1.0. For each trajectory, at most 3 memory items could be extracted. Crucially, we

---

**System Instruction**

You are an expert in evaluating the performance of a web navigation agent. The agent is designed to help a human user navigate a website to complete a task. Given the user's intent, the agent's action history, the final state of the webpage, and the agent's response to the user, your goal is to decide whether the agent's execution is successful or not.

There are three types of tasks:

1. Information seeking: The user wants to obtain certain information from the webpage, such as the information of a product, reviews, map info, comparison of map routes, etc. The bot's response must contain the information the user wants, or explicitly state that the information is not available. Otherwise, e.g. the bot encounters an exception and respond with the error content, the task is considered a failure. Besides, be careful about the sufficiency of the agent's actions. For example, when asked to list the top-searched items in a shop, the agent should order the items by the number of searches, and then return the top items. If the ordering action is missing, the task is likely to fail.

2. Site navigation: The user wants to navigate to a specific page. Carefully examine the bot's action history and the final state of the webpage to determine whether the bot successfully completes the task. No need to consider the bot's response.

3. Content modification: The user wants to modify the content of a webpage or configuration. Carefully examine the bot's action history and the final state of the webpage to determine whether the bot successfully completes the task. No need to consider the bot's response.

*IMPORTANT*
Format your response into two lines as shown below:
Thoughts: <your thoughts and reasoning process>"
Status: "success" or "failure"

**Input Prompt**

**User Intent:** {intent}

**Trajectory:** {trajectory}

**The detailed final state of the webpage:** ```md {cap} ```

**Bot response to the user:** {response if response else "N/A"}

---

Figure 10: System instructions for obtaining binary signals indicating success or failures of the current trajectory.

---

**System Instruction**

You are an expert in web navigation. You will be given a user query and **multiple trajectories showing how an agent attempted the task**. Some trajectories may be successful, and others may have failed.

## Guidelines
Your goal is to **compare and contrast** these trajectories to identify the most useful and generalizable strategies as memory items.
Use **self-contrast reasoning**:
– Identify patterns and strategies that consistently led to success.
– Identify mistakes or inefficiencies from failed trajectories and formulate preventative strategies.
– Prefer strategies that generalize beyond specific pages or exact wording.

## Important notes
- Think first: Why did some trajectories succeed while others failed?
- You can extract *at most 5* memory items from all trajectories combined.
- Do not repeat similar or overlapping items.
- Do not mention specific websites, queries, or string contents — focus on generalizable behaviors and reasoning patterns.
- Make sure each memory item captures **actionable** and **transferable** insights.

## Output Format
Your output must strictly follow the Markdown format shown below:
``` # Memory Item i
## Title <the title of the memory item>
## Description <one sentence summary of the memory item>
## Content <1-5 sentences describing the insights learned to successfully accomplishing the task> ```

**Input Prompt**

**Query:** <user query>

**Trajectories:** <trajectory 1>\n<trajectory 2>\n...<trajectory k>

---

**First-time Check Instruction**

Important: Let's carefully re-examine the previous trajectory, including your reasoning steps and actions taken.

Pay special attention to whether you used the correct elements on the page, and whether your response addresses the user query. If you find inconsistencies, correct them. If everything seems correct, confirm your final answer.

Output must stay in the same "<think>...</think><action></action>" format as previous trajectories.

---

**Follow-up Check Instruction**

Let's check again.

Output must stay in the same "<think>...</think><action></action>" format as previous trajectories.

---

Figure 11: System instructions for memory-aware test-time scaling: the left panel shows parallel scaling (comparing multiple trajectories to extract generalizable insights), while the right panel shows sequential scaling (iteratively re-checking a trajectory to refine the final answer).

induce items from *both* successful and failed trajectories. Successes provide validated strategies, while failures supply counterfactual pitfalls that act as negative signals. To determine success or failure, we adopt an LLM-based binary classifier following (Pan et al., 2024; Wang et al., 2025d). The classifier is prompted with the trajectory and the given user query, and asked to output a categorical judgment (Success or Failure) as shown in Figure 10. Similarly, the backbone of the classifier is set to the same as the agent system, with decoding temperature setting to 0.0 for determinism.

**Memory Retrieval and Response Generation.** For retrieval, we embed each task query using `gemini-embedding-001` (Lee et al., 2025), accessed via Vertex AI.[5] Similarity search is conducted

---

[5] https://ai.google.dev/gemini-api/docs/embeddings

over the memory pool using cosine distance. We select memory items of the top-$k$ most similar experiences (default $k = 1$; ablation study in §5). The retrieved items are concatenated into the agent's system prompt with a simple formatting template (each item represented by its title and content) and instruction:

> Below are some memory items that I accumulated from past interaction from the environment that may be helpful to solve the task. You can use it when you feel it's relevant. In each step, please first explicitly discuss if you want to use each memory item or not, and then take action.

**Memory Consolidation.** After finishing each new query, the trajectory is processed by the extraction pipeline to produce new memory items, which are appended into the memory pool. We adopt a minimal consolidation strategy: newly generated items are directly added without additional pruning. This choice highlights the contribution of REASONINGBANK itself without introducing confounding factors from complex consolidation algorithms. Nevertheless, more advanced consolidation mechanisms (e.g., merging, forgetting) can be incorporated in future work.

**REASONINGBANK Storage** We maintain REASONINGBANK in a JSON format, and each entry of REASONINGBANK consists of a task query, the original trajectory, and the corresponding memory items. All memory items are stored with the schema {title, description, content}. The embedding is pre-computed for each given query and stored in another JSON file for efficient similarity search. We persist the memory pool for each independent run, enabling continual accumulation of experiences throughout test-time learning.

### A.3  MATTS DETAILS

**Prompt Used for MATTS** Figure 11 illustrates the system instructions used in our MATTS framework mentioned in Section 3.3. In the parallel scaling setting (left), multiple trajectories for the same query—both successful and failed—are provided, and the model is instructed to perform self-contrast reasoning. Instead of relying on the LLM to act as an external judge of quality, the model is guided to directly compare and contrast trajectories, identifying patterns that lead to success and mistakes that cause failure. This provides a contrastive signal that grounds the memory extraction process in observable differences between outcomes, yielding more reliable and transferable insights. In the sequential scaling setting (right), the model repeatedly re-examines its own trajectory with check instructions, ensuring consistency and correction over iterations without appealing to external judgment.

**Best-of-N Calculation Details.** Given the task query and $N$ trajectories from the agent system, we leverage an LLM and selects the best answer from the $N$ trajectories. The LLM is initiated as the same backbone LLM as the agent system (e.g., if the agent system uses Gemini-2.5-flash, then the model also uses Gemini-2.5-flash). We feed all the $N$ trajectories to the model at once and use a carefully curated prompt shown in Figure 12, asking the model to select the best answer.

## B  DETAILS FOR EXPERIMENT SETTINGS

### B.1  WEB BROWSING

In this section, we detail the experiment settings used for web browsing agents mentioned in Section 4.1.

**Datasets.** We test REASONINGBANK on three agentic datasets for benchmarking web browsing and coding agents. Specifically, we conduct experiments on WebArena (Zhou et al., 2024) which features general web navigation across diverse domains, spaning shopping, administration, coding (Gitlab), and forums (Reddit). Another benchmark we used is Mind2Web (Deng et al., 2023), which provides playground to test the generalization of agents on versatile operations and environments, including cross-task, cross-website, and cross-domain settings. There are 684 and 1341 test instances in total for WebArena and Mind2Web, respectively. For WebArena, the number of

| **System Instruction** |
| --- |
| You are an expert in evaluating web navigation agent trajectories. You will be given the user query, and {N} candidate trajectories, each representing a sequence of steps for solving the same task. Your job is to select the single best trajectory that most effectively and efficiently solves the task, and explain your reasoning. |

## Input Format:
Each trajectory consists of multiple steps. For each step, you will be provided:
– step_num: Step index in the trajectory.
– action_output: The action the agent takes (click, type, scroll, etc.).
– think_output: The agent's reasoning or plan before taking the action.

## Evaluation Criteria:

### Progress Toward Goal 1. How well the trajectory advances toward completing the user's task. 2. Reward tangible, meaningful progress; penalize minimal or no advancement. 3. Consider both individual step contributions and overall progress.

### Trajectory Efficiency 1. How efficiently the trajectory achieves progress given the number and complexity of steps. 2. Reward significant progress in fewer steps. 3. Favor better value-to-depth ratios. 4. Reward efficient search space exploration.

### Loop Detection: Identify loops or redundant actions. 1. Real Loops: Repeating identical observations and actions with no added value. 2. Benign Repetitions: Slight variations that still yield new information. 3. Penalize real loops heavily; penalize benign repetitions only if they waste effort.

### Error Severity and Stability: Assess severity of errors: 1. Fatal/Blocking: Major penalty. 2. Significant: Moderate penalty. 3. Minor/Recoverable: Minor penalty. 4. Penalize unstable or incoherent model reasoning. 5. Consider whether errors prevent goal completion.

### Overall Trajectory Quality 1. Logical flow of steps, clarity of strategy, and coherence. 2. Balanced exploration vs. exploitation. 3. Closeness to final goal. 4. Reward consistent progress and coherent planning.

## Output Format:
Return the evaluation as a JSON object: ``` { "index": [best_trajectory_index], "analysis": "Detailed reasoning explaining why this trajectory is the best, referencing progress, efficiency, loop detection, error severity, and overall quality." } ```

| **Input Prompt** |
| --- |

**Query:** {query}

**Trajectory 1:** {trajectory_1}\n**Trajectory 2:** {trajectory_2}\n……\n**Trajectory N:** {trajectory_N}

Figure 12: System instructions for obtaining the best answer from $N$ candidate trajectories for BoN calculation.

instances for different domains are Shopping (187), Admin (182), Gitlab (180), Reddit (106), and Multi (29). For Mind2Web, the number of different settings are Cross-Task (252), Cross-Website (177), and Cross-Domain (912).

**Baselines.** We compare REASONINGBANK against several representative memory-augmented approaches: (i) *Vanilla*, the backbone LLM agent without any memory module, serving as a reference point; (ii) *Synapse* (Zheng et al., 2024), a representative work that organizes past trajectories as in-context memory; and (iii) *AWM* (Wang et al., 2025d), which further abstracts common patterns from trajectories into reusable workflows. Together, these baselines span a progression from agents without memory, to those that directly reuse past trajectories, and finally to methods that distill higher-level structures, providing a comprehensive comparison for evaluating REASONINGBANK. To ensure a fair comparison, the baselines are implemented with the same "Memory Retrieval" and "Memory Consolidation" mechanisms. The only difference is about "Memory Extraction", which is exactly how REASONINGBANK different from baselines in terms of memory formulations.

**Implementation Details.** We build our agents upon several state-of-the-art LLMs accessed via the Vertex AI API,[6] including Gemini-2.5-Flash, Gemini-2.5-Pro (Comanici et al., 2025), and Claude-3.7-Sonnet (Anthropic, 2025). These choices allow us to investigate both cross-family (Gemini, Claude) and intra-family (Flash, Pro) variations. BrowserGym (de Chezelles et al., 2025) is used as the execution environment for WebArena, where we set a maximum step limit of 30 per query. The agent is implemented in ReAct (Yao et al., 2023) style, and iterates until the model predicts the stop action or reaches a task termination condition. We use the decoding temperature of $0.7$ for model generations for both WebArena and Mind2Web.

**Evaluation Metrics.** For WebArena benchmark, we evaluate all methods across two key dimensions: *effectiveness* and *efficiency*. For effectiveness, we report the *success rate (SR)*. A task is marked as "successful" only if the agent's final output or state precisely matches the pre-defined ground-truth goal, which is measured by the evaluation protocol from the corresponding benchmarks. SR is the total number of successful tasks divided by the total number of tasks evaluated, formally, it is calculated as $SR = \frac{1}{N} \sum_{i=1}^{N} \text{isSuccess}(q_i)$, where $\text{isSuccess}(q_i)$ is the binary function that returns 1 if task $q_i$ is successful and 0 otherwise. For efficiency, we measure

---

[6] https://cloud.google.com/vertex-ai

the *average number of steps (AS)* taken by the agent to complete each query, which reflects the computational and interaction cost incurred during task completion. A single step is defined as one complete agent-env interaction cycle following the ReAct loop, which typically involves observing the current state, generating a thought, and a subsequent action. AS is calculated as the total number of steps taken in the trajectory when solving task $q_i$ divided by the total number of tasks, specifically, $AS = \frac{1}{N}\sum_{i=1}^{N} \text{Steps}(q_i)$. For Mind2Web dataset, each task in has a predefined fixed number of steps; at each step, the agent needs to predict an action, which is evaluated by: *element accuracy*: to check if the correct page element is selected, *action F1* to check if the action taken on the element is correct. Aggregating element accuracy and action F1 yields *step success rate* which checks that both element and action selection are correct at the current step. Lastly, after completing every step in the given task, the last metric *task-level success rate* measures if all intermediate steps are successfully conducted for this task, i.e., all steps for this task score 1.0 under metric step success rate.

## B.2 SOFTWARE ENGINEERING

**Dataset.** To benchmark agentic coding tasks, we evaluate on SWE-Bench-Verified (Jimenez et al., 2024), a repository-level issue resolution benchmark. The dataset consists of 500 high-quality test instances that have been manually verified. Each instance requires generating a patch to address the underlying bug described in the input issue. The objective is to modify the relevant portions of the codebase such that all provided test scripts execute successfully.

**Metrics.** We report the issue resolution rate on SWE-Bench-Verified as the primary evaluation metric. The resolution rate measures the percentage of issues successfully fixed across all data points, where an issue is deemed resolved if the submitted patch passes all test scripts. To evaluate the patch application rate, we attempt to apply the generated patches to the repository using the standard `patch` program, counting only successful applications. Our implementation follows the official evaluation scripts.[7] For efficiency, we additionally report the average number of steps performed by the agent per instance, following web-browsing tasks.

**Implementation.** We implement REASONINGBANK for SWE-Bench following the setting of mini-SWE-Agent (Yang et al., 2024), which enforces the Bash-Only environment with no tools and no special scaffold structure. It assumes a simple ReAct agent loop (Yao et al., 2023). Similar to previous experiments, we compare REASONINGBANK against (i) No memory and (ii) Synapse. [8]

## C ADDITIONAL ANALYSES

### C.1 NUMBER OF RETRIEVED EXPERIENCES

We conduct another ablation study on different number of retrieved experiences using Gemini-2.5-flash on WebArena-Shopping subset. As shown in Figure 13, we found that incorporating relevant memory significantly boosts performance (from 39.0 without memory to 49.7 with one experience). However, as the number of experiences increases, the success rate gradually declines (46.0 with 2, 45.5 with 3, and 44.4 with 4). This suggests that while memory provides valuable guidance, excessive experiences may introduce conflicts or noise. Hence, the relevance and quality of memory are more crucial than sheer quantity for effective performance.

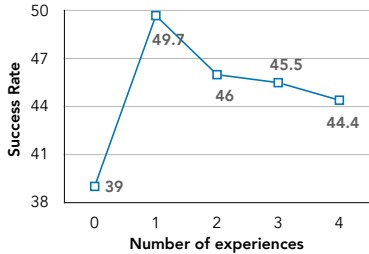

Figure 13: Ablation results for using various number of experiences.

---

[7]https://www.swebench.com/SWE-bench/api/harness/

[8]We exclude AWM here because the action space in mini-SWE-Agent is open-ended (arbitrary Bash commands), making it difficult to extract the common routines or fixed workflows that AWM requires for cross-task generalization.

## C.2 INFERENCE COST STUDY

In this section, we provide a comprehensive view on inference cost for REASONINGBANK and baselines to facilitate real-world deployment. We report a breakdown of the averaged total token consumption for each trajectory in addition to number of interaction steps mentioned in Section 4.1. The results are shown in Table 5.

Table 5: Breakdown results of total token consumption required for each task.

| Methods | Action Generation | LLM-as-a-Judge | Memory Extraction | Total |
|---|---|---|---|---|
| No Memory | 50847.4 | - | - | 50847.4 |
| Synapse | 55920.5 | 2594.2 | - | 58514.7 |
| AWM | 53819.6 | 2479.1 | 3074.1 | 59372.8 |
| REASONINGBANK | 49306.1 | 2186.3 | 1562.1 | 53054.5 |

From the table we can see that compared with "No memory", while the total token consumption is increased only by around 4.3%, the overall performance is boosted by 20.5%. Other memory baselines such as Synapse and AWM will greatly increase the computation overhead while achieving less performance gains compared with REASONINGBANK, demonstrating the cost-effectiveness of REASONINGBANK.

## C.3 PASS@k ANALYSIS

**Memory-aware scaling improves sample efficiency and sustains stronger performance gains.** Pass@$k$ analysis under parallel scaling on WebArena-Shopping subset with Gemini-2.5-flash (Figure 14) reveals two distinct effects. First, MATTS w/o aggregation (Vanilla TTS) already makes test-time learning behave similarly to RL training: instead of inflating pass@$k$ at large $k$, it improves sample efficiency by guiding exploration. For example, at $k = 2$, MATTS w/o aggregation achieves 50.8 compared to 47.6 from MATTS w/o memory, extracting more value from each rollout as noted in (Yue et al., 2025). Second, equipping TTS with memory-aware scaling pushes performance further. MATTS not only preserves efficiency at small $k$ (51.3 at $k = 2$) but also sustains strong growth with scaling, reaching 62.1 at $k = 5$, compared to only 52.4 for MATTS w/o memory. Overall, we see that MATTS unlocks more potential of agent systems and encourages diversified generation for better pass@$k$ performance.

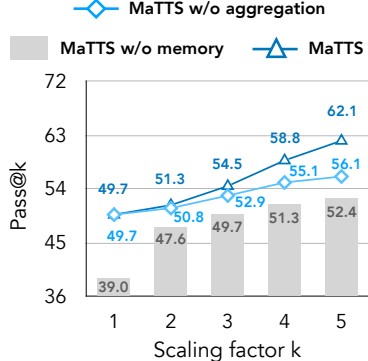

Figure 14: Pass@$k$ under parallel scaling with REASONINGBANK.

## C.4 RESULTS WITH SMALLER OPEN-SOURCE LLMs

To evaluate the generalizability of REASONINGBANK across different model scales and architectures, we extend our evaluation to smaller, publicly available open-source models. Specifically, we conduct experiments using Gemma-3-12B-Instruct[9] on the WebArena-Shopping subset. As summarized in Table 6, REASONINGBANK continues to achieve consistent performance gains even when deployed on more compact models. While the baseline success rate without memory is 17.1%, REASONINGBANK improves this to 24.1%, outperforming other memory-based methods such as Synapse (16.0%) and AWM (21.4%). Furthermore, REASONINGBANK demonstrates superior efficiency by requiring the fewest average interaction steps (11.8 steps) compared to all baselines. These results suggest that the benefits of our memory induction approach are not limited to proprietary frontier models but effectively scale down to smaller open-source LLMs.

## C.5 CASE STUDY

To better illustrate the benefits of our approach, we present three representative case studies.

---

[9] https://huggingface.co/google/gemma-3-12b-it

Table 6: Performance on WebArena-Shopping using Gemma-3-12B-Instruct.

|                   | No memory | Synapse | AWM | REASONINGBANK |
|-------------------|-----------|---------|------|---------------|
| Success Rate (%)  | 17.1      | 16.0    | 21.4 | **24.1**      |
| Average Steps     | 13.7      | 14.0    | 12.5 | **11.8**      |

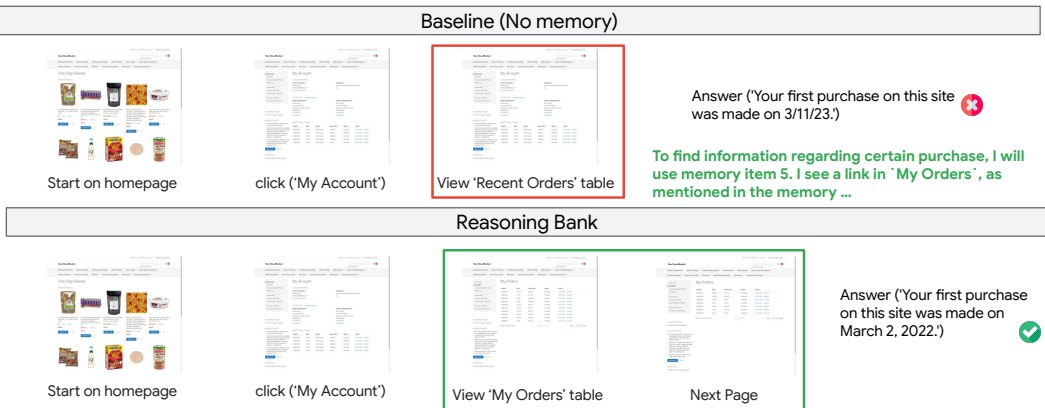

Figure 15: REASONINGBANK enables the agent to recall and apply past reasoning hints, guiding it to the full order history and yielding the correct first purchase date, unlike the baseline that fails with only recent orders.

**Effectiveness.** Figure 15 highlights the effectiveness of REASONINGBANK in leveraging related previous experiences as memory items. While the baseline agent (without memory) only checks the "Recent Orders" table and mistakenly outputs the most recent purchase date, REASONINGBANK recalls from past reasoning hints to explore the full purchase history and correctly identifies the earliest order.

**Efficiency.** Figure 16 demonstrates the efficiency gains. In a navigation-heavy shopping task, the baseline requires 29 steps due to repeated inefficient browsing. It stucks and struggles to find the correct place of filter for "Men". In contrast, REASONINGBANK leverages stored reasoning about category filtering, enabling the agent to directly reach the relevant items and complete the task in only 10 steps.

**Reflection on Successes and Failures.** Figure 17 shows how memory items induced by REASONINGBANK through reflecting on past trajectories helps to prevent similar errors from happening again, which enables emergent improvement. In this case, the original trajectory actually fails because of the imprecise search query that leads to numerous returned items, and irrelevant objects. REASONINGBANK is able to first reflect on the trajectory, pinpoint the key reason of failure, and extract valuable strategies that would avoid similar errors such as search query optimization and using the functionality filters.

## D  FUTURE DIRECTIONS

In this section, we briefly discuss the potential future directions following REASONINGBANK and MATTS.

**Modular and Compositional Memory.** Our current framework distills each experience into multiple memory items, and when a new query arrives, we retrieve similar experiences and reuse all associated items independently. This design highlights the effect of memory content but does not consider how items could be composed into higher-level strategies. Future work could

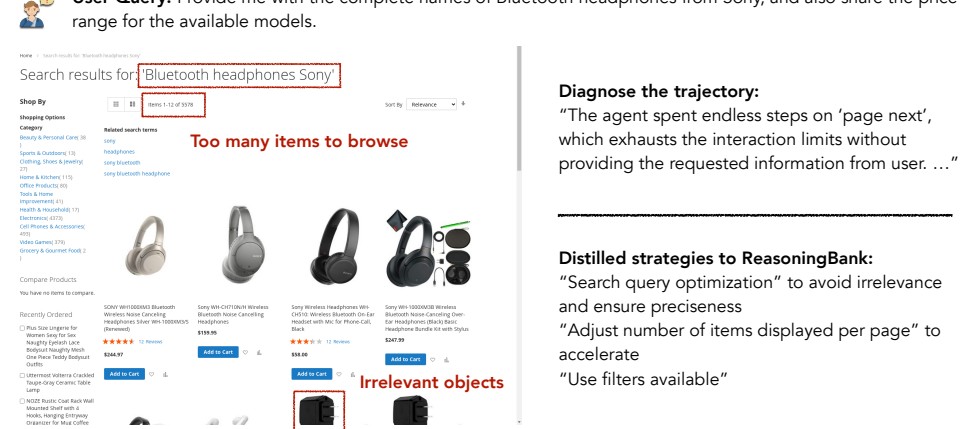

Figure 16: REASONINGBANK improves efficiency by leveraging past reasoning hints, reducing the navigation from 29 steps to 10 steps compared to the baseline without memory.

Figure 17: Memory items induced by REASONINGBANK unlocks emergent improvement of REASONINGBANK, which help avoid similar errors from happening again.

explore composition-aware retrieval and consolidation with based on modular memory extraction, enabling the agent to combine complementary items or form reusable macros, thereby yielding richer strategies and stronger generalization in long-horizon tasks. For example, memory could be extracted with respect to "planning memory", "tool-use memory", "operational memory", "user-centric memory", etc. In this way, memory extracted would be more fine-grained and memory retrieval could unlock compositional and complementary power, not just task similarity.

**Advanced Memory Architectures.** Our current system design is intentionally minimal; a natural next step is to build a layered, product-level memory stack that integrates mature paradigms — e.g., episodic traces (Fountas et al., 2025) for per-task context, short-term "working" memory (Lumer et al., 2025) for within-session state, and long-term (Wang et al., 2025b) consolidated knowledge with decay/refresh policies. The philosophy of REASONINGBANK are compatible with the above different memory angularities. Additionally, the current memory retrieval could also move beyond embedding-based similarities to reasoning-intensive controllers (Shao et al., 2025) that decompose queries, plan multi-hop lookups across tiers, and condition selection on uncertainty, recency, and cost. Learning-based routers and consolidation policies could also automate this process. This integration would turn REASONINGBANK with MATTS into a deployable memory service that scales across domains and teams.

# E    LIMITATIONS

While REASONINGBANK demonstrates strong empirical performance and introduces a practical paradigm for memory as a scaling dimension, it also comes with several limitations that suggest directions for future research.

**Focus on memory content.** Our study emphasizes how to curate and utilize memory content (e.g., integrating failure trajectories, constructing distilled reasoning cues). For this reason, we did not extensively compare with other memory architectures such as episodic or hierarchical memory. These designs address orthogonal concerns (memory form/structure), while our contribution targets what should be stored and reused. Exploring their combination would be an interesting future direction.

**Simplicity in memory retrieval and consolidation.** We intentionally adopt simple embedding-based retrieval and straightforward consolidation to better isolate the effect of content quality. More sophisticated strategies (e.g., adaptive retrieval, hierarchical consolidation) are compatible with our framework and could further enhance performance, but are not the focus of this work. This choice ensures that the observed gains can be attributed directly to the design of reasoning-oriented memory content.

**Dependence on LLM-as-a-judge for correctness signals.** In our implementation, success and failure signals for trajectories are determined by an LLM-as-a-judge. While this automatic labeling enables scalable evaluation without ground-truth feedback, it may introduce noise when tasks are ambiguous or when the judge model itself errs. While our results suggest the framework remains robust under such noise, future work could incorporate stronger verifiers, human-in-the-loop feedback, or ensemble judgment to enhance the reliability of memory induction.

# F    USE OF LLMS

We used LLMs as a general-purpose writing assist tool during the preparation of this submission. Specifically, LLMs were employed for polishing the clarity and readability of text (e.g., refining sentence structure, improving grammar, and shortening overly verbose phrasing). All research ideas, methodology design, experiments, analyses, and final writing decisions were conceived, implemented, and validated solely by the authors.

