# OpenReview forum: "ReasoningBank: Scaling Agent Self-Evolving with Reasoning Memory"
_ICLR.cc/2026/Conference — ICLR 2026 Poster_

### Official Review · Reviewer_sUDi · 2025-10-29

**Soundness:** 3
**Presentation:** 3
**Contribution:** 2
**Rating:** 6
**Confidence:** 5

**Summary:**

This work addresses a key limitation of large language model agents: their inability to learn from accumulated experience in ongoing tasks like web browsing and software engineering. The authors propose two innovations. REASONINGBANK is a structured memory that extracts generalizable reasoning strategies from both successes and failures, operating in a loop of retrieval, construction, and integration. MATTS combines REASONINGBANK with test-time expansion: memory guides exploration while diverse expansions enrich memory and improve performance. The main contributions are a memory framework that leverages failures for generalizable reasoning, a synergy with test-time expansion, and empirical validation of memory-driven agent self-improvement.

**Strengths:**

1. The problem definition is novel, focusing on extracting reasoning from experience rather than mere experience storage, and highlighting the value of failed experiences.
2. The experiments are extensive, with diverse benchmarks and thorough ablations, providing strong evidence for the effectiveness of REASONINGBANK.

**Weaknesses:**

1. The paper evaluates generalization only within subsets of WebArena and Mind2Web. The claimed generalization gains may simply reflect similarities between tasks within the same benchmark, as human or LLM patterns are often consistent internally. Stronger evidence would require cross-benchmark tests, such as using WebArena Shopping as an experience pool to evaluate completely unrelated Mind2Web tasks, or testing SWE-Bench with WebArena experiences.
2. Success and failure of trajectories are judged by the LLM, but the accuracy of these judgments is not analyzed. Using the LLM as a critic may propagate errors.
3. Experiments focus only on historical task-level memory, without considering temporal or user-level memory, which are more valuable in practice. Task-level memory yields only modest metric gains that could also be achieved with additional SFT data or RL, but at the cost of efficiency, storage, and increased framework complexity. Temporal or user-level memory cannot be solved as easily and would provide more significant practical value, yet these aspects are not studied.
4. The analysis of emergent behavior is limited to changes in the memory bank. It does not examine whether prior failed tasks can be corrected through reflection on successes and failures, which would demonstrate true emergent capability. Currently, the memory module mainly helps stabilize model outputs via complex multi-step processes, increasing engineering complexity, lowering scalability, and reducing efficiency. SFT or RL may achieve similar gains more efficiently. Showing experiments that capture emergent improvement would significantly increase the work’s impact.
5. Only strong closed-source models are evaluated. Open-source models and models of different sizes are not tested.
6. Experiments are limited to web navigation and software engineering agents, with insufficient evaluation on more general tool-use agents such as Taubench or BFCL. Moreover, the experience extraction prompt appears tailored to web navigation, implying that each new scenario may require a custom prompt and tuning, reducing usability and scalability.
7. While the paper claims efficiency gains in terms of fewer steps, actual runtime may be higher. The memory module adds significant inference overhead, particularly on the TTFT, which may degrade user experience and hinder practical deployment in real-world agent scenarios.

**Questions:**

1. Has any subset of trajectories been manually labeled to verify the consistency between LLM Critic judgments and human labels? If there is a discrepancy (e.g., LLM misclassifies 15% of failed trajectories as successful), how would this affect the quality of REASONINGBANK memories and downstream agent performance?
2. How does REASONINGBANK perform when the memory bank grows large (e.g., over 500 entries)? Could pruning or merging strategies, such as removing low-similarity memories relative to the current query, maintain or improve performance? Which metrics are used to evaluate the effect of such optimizations?
3. In sequential expansion, how is the termination condition determined (e.g., fixed k=5 steps versus stopping when no new insights emerge)? Do intermediate optimization notes (e.g., “recheck filtering labels”) provide unique value to memory, or are they largely redundant compared with final trajectory insights?
4. Can memory entries from web browsing tasks transfer to software engineering tasks (e.g., using a “web form data consistency check” strategy for code debugging)? If not, how can the framework be adapted to support cross-task memory transfer?
5. Are failures categorized by type (e.g., “operational errors” versus “reasoning errors,” such as clicking the wrong button versus misinterpreting a user query), and are separate insights extracted for each type? Would such granularity improve the practical usefulness of the memory?

---

> ### Author Response · Authors · 2025-11-18
> **Response to Reviewer sUDi (Part I)**
>
> Thanks for your detailed review and constructive comments. We appreciate your recognition of our novel problem definition and extensive experiments that support the effectiveness of ReasoningBank. Below, we provide detailed responses to each of your comments and hope to address any further considerations you may have.
>
> ---
>
> > **W1: Generalization is only tested between similar tasks within the same benchmark. Stronger evidence would require cross-benchmark tests.**
>
> Thanks for your comment. We would like to first clarify our experiment setting on generalization. In Tables 1 and 3, we report the results on multiple websites and cross-domain results for WebArena/Mind2Web. The tasks are not similar, because they are built upon quite different websites with completely different layouts and functionalities. Performing consistently well compared with degraded baselines, ReasoningBank already demonstrates its generalization capability for web-browsing tasks by leveraging experiences from one website/domain category to improve performance on another distinct one.
>
> Additionally, we are not claiming ReasoningBank as a method for transfer learning or a tool specifically designed for “generalization on completely unrelated tasks”. Our general assumption is “agents learn from past useful experiences” as introduced in Lines 40ff. That being said, we humbly think that the intended use case of ReasoningBank misaligns with your expectation on “cross-benchmark generalization”.
>
> Therefore, while cross-benchmark generalization is good for a memory system (and we show that ReasoningBank generalizes to some extent), it is not a must-have in our setting and we are not claiming so.
>
> ----
>
> > **W2 & Q1: Success and failure of trajectories are judged by the LLM, but the accuracy of these judgments is not analyzed. Using the LLM as a critic may propagate errors.**
>
> Thanks for your constructive comment. Following your suggestion, we additionally conduct experiments to study the calibration and robustness on the self-verification step. Following our analysis setting in Section 5, we run experiments with gemini-2.5-flash on the WebArena-Shopping subset.
>
> We first calculated the accuracy for the model’s own self-assessment (LLM-as-a-Judge) using the ground-truth label, which turns out to be 72.7%. To systematically study the robustness/sensitivity of this step, we simulate different accuracies for the self-assessment step from the perfect 100% (ground-truth) down to the worst-case 50% (random-guessing). Detailed descriptions of the setting could be found in **Appendix C. 2 of the revised PDF**. Overall results are presented in the following table:
>
> | | 100% (ground-truth) | 90% | 80% | 70% | 60% | 50% (random-gussing) |
> | :--- | :--- | :--- | :--- | :--- | :--- | :--- |
> | **ReasoningBank** | 52.4 | 49.7 | 48.7 | 49.7 | 47.6 | 47.6 |
>
> The results clearly show that the performance of ReasoningBank is highly robust to variations in the self-verification accuracy. The overall accuracy **remains largely stable** within the practical range of $70\%-90\%$ self-judgement accuracy, demonstrating that our approach is not overly sensitive to the quality of the self-judgement signal. This robustness makes ReasoningBank broadly applicable even to tasks where self-verification is challenging or provides non-binary signals.
>
> Additionally, we want to emphasize that the step of explicit self-verification/judgement can be **completely bypassed** using MATTS with the principle of leveraging self-contrastive signals from multiple trajectories while maintaining huge performance improvement.

---

> ### Author Response · Authors · 2025-11-18
> **Response to Reviewer sUDi (Part II)**
>
> > **W3: Experiments focus only on historical task-level memory, without considering temporal or user-level memory, which are more valuable in practice. Task-level memory yields only modest metric gains that could also be achieved with additional SFT data or RL, but at the cost of efficiency, storage, and increased framework complexity. Temporal or user-level memory cannot be solved as easily and would provide more significant practical value, yet these aspects are not studied.**
>
> Thanks for your feedback. As stated in Section 2, the central goal of ReasoningBank is to investigate test-time improvement: how an agent can enhance its performance on future tasks by accessing past experiences. In this case, task-level memory is the central memory type we focus on. This is in distinct contrast to other applications of agent memory introduced as personalization and long-context management (line 100), where memory types such as temporal-level memory (for long-context management) and user-level memory (for personalization) would dominate. Therefore, we humbly believe that while temporal-level memory and user-level memory hold important value in agent memory research, they are out-of-scope for this specific paper.
>
> Additionally, we want to argue that the performance gains brought by ReasoningBank with task-level memory is not modest, but significant. Compared to baselines, we achieved a relative improvement of 24.1% at most. SFT/RL can also increase performance, but at the cost of high-quality data, huge training efforts, and decreased generalization. The purpose of ReasoningBank is also not to replace SFT/RL. The memory formulation we established here could be combined and serve as memory recipes for memory-aware agent training. For example, we can periodically convert the memory into high-quality training data.
>
>
> ----
>
> > **W4: The analysis of emergent behavior is limited to changes in the memory bank. It does not examine whether prior failed tasks can be corrected through reflection on successes and failures, which would demonstrate true emergent capability.**
>
> Thanks for your constructive comment. In order to demonstrate the emergent capability of ReasoningBank, we provide an additional case study in **Appendix C.3 "Emergent Capabilities"**. In this specific example, we can see that the original trajectory fails because of the imprecise search query that leads to numerous returned items, and irrelevant objects. The agent ultimately exhausted in browsing all the items returned, and exceeding interaction steps of 30. ReasoningBank is able to first reflect on the trajectory, pinpoint the key reason of failure, and extract valuable strategies that would avoid similar errors such as search query optimization and using the functionality filters.
>
> ----
>
> > **W5: Only strong closed-source models are evaluated. Open-source models and models of different sizes are not tested.**
>
> Thanks for your insightful suggestion. Our existing experiments focus on closed-source models of different model families (gemini, claude) and sizes (gemini-2.5-flash, gemini-2.5-pro). We originally did not focus on open-source models because the benchmarks are pretty challenging for small open-source models.
>
> We agree with you that testing on smaller, publicly available models would further strengthen our results. Therefore, we conducted experiments using Gemma-3-12B-Instruct (https://huggingface.co/google/gemma-3-12b-it) on the WebArena-Shopping subset. Results are shown in the following table:
>
> | | No memory | Synapse | AWM | ReasoningBank |
> | :--- | :--- | :--- | :--- | :--- |
> | **Success Rate** | 17.1 | 16.0 | 21.4 | 24.1 |
> | **Average Steps** | 13.7 | 14.0 | 12.5 | 11.8 |
>
> We can observe from the results that ReasoningBank still achieves consistent performance gains with much less average interaction steps.

---

> ### Author Response · Authors · 2025-11-18
> **Response to Reviewer sUDi (Part III)**
>
> > **W6: Experiments are limited to web navigation and software engineering agents, with insufficient evaluation on more general tool-use agents such as Taubench or BFCL. Each new scenario may require a custom prompt and tuning, reducing usability and scalability.**
>
> Thanks for your feedback. Our current experiments chose web navigation and software engineering as they represent highly challenging and impactful domains, and are widely adopted by agent frameworks for evaluation, which makes our life easier for making direct comparison with baseline methods. We agree that evaluation on broader tool benchmarks like Taubench or BFCL is interesting. We also believe that by making slight changes in memory extraction prompts would easily adapt ReasoningBank to corresponding scenarios since the action space of agent policy for WebArena and SWE-Bench is more complex than synthetic tool sets of either Taubench or BFCL.
>
> Additionally, we wish to clarify that our method does not require heavy prompt and tuning. In fact, as you can observe from Figure 8, only the first sentence of prompt is related to the specific scenario, which could be easily modified for new settings.
>
> ----
>
> > **W7: While the paper claims efficiency gains in terms of fewer steps, actual runtime may be higher. The memory module adds significant inference overhead, particularly on the TTFT, which may degrade user experience and hinder practical deployment in real-world agent scenarios.**
>
> Thanks for your constructive comment and for recognizing ReasoningBank’s efficiency in terms of fewer interaction steps. To make the breakdown clearer for inference cost, we additionally conducted experiments on calculating actual tokens consumed including both input and output by each method in the following table in terms of different stages (averaged token number for each instance in WebArena-Shopping tasks):
>
> | | Action generation | LLM-as-a-Judge | Memory Extraction | Total |
> | :--- | :--- | :--- | :--- | :--- |
> | **No memory** | 50847.4 | - | - | 50847.4 |
> | **Synapse** | 55920.5 | 2594.2 | - | 58514.7 |
> | **AWM** | 53819.6 | 2479.1 | 3074.1 | 59372.8 |
> | **ReasoningBank** | 49306.1 | 2186.3 | 1562.1 | 53054.5 |
>
> From the table we can see that compared with “No memory”, while the total token consumption is **increased only by around 4.3%**, the overall **performance is boosted by 20.5%**. Other memory baselines such as Synapse and AWM will greatly increase the computation overhead while achieving less performance gains compared with ReasoningBank, demonstrating the cost-effectiveness of ReasoningBank.
> We have updated the new results and corresponding analyses in **Appendix C.3**.
>
> We also compute the actual inference time for each method per task for WebArena-Shopping subset using Gemini-2.5-flash in the following table:
>
> | | No memory | Synapse | AWM | ReasoningBank |
> | :--- | :--- | :--- | :--- | :--- |
> | Inference time | 87.6s | 95.3s | 101.4s | 104.9s |
>
> We can see that with our current implementation, ReasoningBank incurs around 19% inference latency while improving the performance by 27%, which is cost-effective. Meanwhile, we argue that our implementation could be further improved by engineering efforts when deployed in real-world applications. For example, the actual inference time could further be enhanced by memory extraction in the background (sleep-time compute [1], compared to implementation as blocking operation) and MIPS for memory retrieval.
>
> [1] Lin, Kevin, et al. "Sleep-time compute: Beyond inference scaling at test-time." arXiv preprint arXiv:2504.13171 (2025).
>
> ----
>
> > **Q2: How does REASONINGBANK perform when the memory bank grows large (e.g., over 500 entries)? Could pruning or merging strategies, such as removing low-similarity memories relative to the current query, maintain or improve performance? Which metrics are used to evaluate the effect of such optimizations?**
>
> Thanks for your question. When ReasoningBank grows large, the bottleneck that may hinder the performance would be the preciseness of memory retrieval, i.e., whether the most relevant/useful experiences could be leveraged. Currently, we select the experiences using the embedding-based search. We appreciate your suggestions on pruning or merging strategies, which could improve the retrieval efficiency and accuracy as well specifically in a large-scale scenario. We can prune memory items regarding the temporal relation or frequency for utility, or merge memory items with respect to certain aspects, e.g., topics or keywords. For evaluation metrics, we can consider retrieval  accuracy (hit rate), diversity/coverage of ReasoningBank, or intrinsic quality evaluation (by human evaluation).

---

> ### Author Response · Authors · 2025-11-18
> **Response to Reviewer sUDi (Part IV)**
>
> > **Q3: In sequential expansion, how is the termination condition determined (e.g., fixed k=5 steps versus stopping when no new insights emerge)? Do intermediate optimization notes (e.g., “recheck filtering labels”) provide unique value to memory, or are they largely redundant compared with final trajectory insights?**
>
> Thanks for your question. In our current implementation, the sequential scaling process is determined by a fixed scaling factor, i.e., fixed k=5 steps here in your comment. Those intermediate optimization notes indeed provide unique value to the memory. Performance-wise, curating memories from sequential scaling is better than vanilla scaling performance according to Figure 4(b). We also examine by case study that intermediate notes in sequential expansion helps to identify, for example, “when and how potential failures could happen”. Consider the user query in Figure 15, the initial trajectory returns incorrect results by navigation wrongly to ``recent orders’’. Upon self-refinement, the agent will generate something like “My apologies. I jump to the conclusion prematurely. The task actually requires me to find the earliest purchase, therefore I should click button [1417] to view all orders …” The final trajectory insights will provide correct navigation to find the required purchase, but the intermediate notes endow curated memory items with high-level navigation strategies such as  “examining all navigation functionalities available”.
>
> ----
>
> > **Q4: Can memory entries from web browsing tasks transfer to software engineering tasks (e.g., using a “web form data consistency check” strategy for code debugging)? If not, how can the framework be adapted to support cross-task memory transfer?**
>
> Thanks for your question. We do observe some memory items induced from web-browsing that might also be useful for software engineering tasks. For example, one memory item is ``validate state after actions and errors’’, which requires the agent to fully examine the current state (e.g., by re-evaluating the AXTree of the page) to confirm the actual action’s effect or to understand the actual state, rather than making blind actions. This could also be transferred to software engineering tasks when a new bash command is performed.
>
> However, similar to W1, we argue that ReasoningBank is not designed as a method for transfer learning or a tool specifically designed for cross-benchmark generalization, and our focus lies more in “agents learn from past useful experiences”.
>
> To better support such cross-task memory transfer, the framework could be adapted in several potential ways: (i) offline population with experiences from a diverse set of tasks to obtain relevant memory items, and (ii) controllable generation of memory items conditioned on domain priors.
>
> > **Q5: Are failures categorized by type (e.g., “operational errors” versus “reasoning errors,” such as clicking the wrong button versus misinterpreting a user query), and are separate insights extracted for each type? Would such granularity improve the practical usefulness of the memory?**
>
> Thanks for your insightful question. We do not explicitly categorize failures and ask memory extractor to extract separate insights. As illustrated in Figure 8 Page 16, memory extractor will first reflect on why the trajectory is successful or failed. During this process, it implicitly demonstrates some categorization of failures, such as “the trajectory spends endless steps in `page next`, which exhausted the interaction steps”, and then curates corresponding memory items for avoidance. In light of this, we think that adding such granularity for error categorization could improve the controllability and compositionality of memory items. Similar principles are explored in Agent-KB [1] and proved to be effective. We further **extended our discussion in Appendix D** “Modular and Compositional Memory” regarding this point.
>
> [1] Tang, Xiangru, et al. "Agent kb: Leveraging cross-domain experience for agentic problem solving." arXiv preprint arXiv:2507.06229 (2025).

---

> > ### Comment · Reviewer_sUDi · 2025-11-26
> >
> > Many thanks to the authors for the detailed response and the additional experiments, which provide convincing results. I believe that ReasoningBank offers very strong empirical results and analyses in the area of agent memory and experience. However, as the authors acknowledge in their rebuttal, ReasoningBank still has several limitations, and, in my view, its main contribution lies in performance improvements rather than a major methodological breakthrough. I will therefore keep my overall rating unchanged, raise my scores for soundness and presentation, and advocate for acceptance.

---

> > > ### Author Response · Authors · 2025-11-27
> > >
> > > Thank you for reading our rebuttal. We are glad to hear that our response and additional experiments provided convincing results, and thank you for raising the scores for both soundness and presentation, and for advocating acceptance of our paper.
> > >
> > > We greatly appreciate your acknowledgment of ReasoningBank's "very strong empirical results and analyses in the area of agent memory and experience", and we also would like to further clarify our core contribution here:
> > > - We formulated an effective and efficient recipe (learning from both successful and failed experiences; leveraging reasoning memory) for agent memory. While the ingredients are not entirely new, the recipe itself and the implementation (integration with agent systems) are sufficiently original as also noted by Reviewer `WYBc` .
> > > - Building upon the recipe, we proposed MATTS, aiming to establish experience-driven memory as another scaling dimension of agents by leveraging the bidirectional synergy between memory and test-time scaling.
> > >
> > > We hope the above clarification reinforces the systemic methodological significance of our work alongside its strong empirical results. Thank you again for your time and effort in helping us to make ReasoningBank better!

---

### Official Review · Reviewer_WYBc · 2025-10-29

**Soundness:** 3
**Presentation:** 4
**Contribution:** 3
**Rating:** 8
**Confidence:** 4

**Summary:**

The paper presents a memory framework for agents called REASONINGBANK and a complementary memory-aware test-time scaling (MATTS) technique. Unlike previous agent memory frameworks, REASONINGBANK (a) extracts generalized learnings (so-called memory items) from previous reasoning trajectories (i.e., attempts at solving a problem) instead of simply memorizing the entire raw trajectories and (b) learns by comparing successful and unsuccessful trajectories instead of only learning from the successful ones. MATTS complements REASONINGBANK by generating multiple reasoning trajectories, either in sequence or in parallel and extracting memory items from across all trajectories. The authors empirically demonstrate that REASONINGBANK substantially outperforms other agent memory frameworks and a baseline without memory in both task-specific success rate as well as efficiency (measured by the number of reasoning steps). They further show that REASONINGBANK and MATTS work particularly well in combination.

**Strengths:**

Originality: While the key ideas (e.g., generalizing learnings, incorporating learnings into prompts, multiple reasoning trajectories, etc.) behind REASONINGBANK and MATTS are not entirely novel and have also been used in similar papers in the last 2 years, the exact way how these techniques are put together and implemented is sufficiently original and this apparently leads to a substantial outperformance of existing approaches.

Quality: The implementation details (e.g., prompts) presented in the main part and the appendix appear very well-crafted. The chosen evaluation experiments are rigorous and suitable to test various aspects of the presented methods.

Clarity: The vast majority of the paper is very clearly presented in both text and visuals. The paper offers easily accessible explanations to all parts of the framework and makes suitable references to the appendix which includes an even more comprehensive documentation. Key ideas (e.g., learning from successful and failed experiences alike) are being repeated in several places, because they are so important.

Significance: The paper presents a well-performing solution to a timely and relevant problem. It further makes extensive references to recent related work and similar techniques. The advantage of the presented methods over their baselines is so substantial and consistent that it can hardly be ignored.

**Weaknesses:**

In some areas the paper could have been more precise or elaborate with some explanations, e.g.,
- The REASONINGBANK framework includes a memory consolidation step. While the name suggests that memories are actually being consolidated (i.e., transforming a set of memories into a somewhat more information-dense, better structured, or simply smaller set of memories), the used implementation does nothing of that but simply appends new memory items to the set of existing ones. This appears like a waste of potential. However, the authors offer a justification for their choice in the appendix (lines 927ff.) arguing that advanced consolidation mechanisms could have led to unwanted confounding factors, and leaving such mechanisms for future work. For their paper, the authors may want to rename “memory consolidation” into something that doesn’t use the word consolidation (which suggests that something more complex is going on) and/or making clearer why they decided to keep their implementation simple in the main part, not just the appendix.
- Line 260 mentions the two main performance metrics “success rate” and “average steps”. While the readers may already have a good intuition about these metrics, it could have been even more clear if the authors had given a more precise definition of how the success rate is calculated and what constitutes a single step.
- The concept of “rollouts” is mentioned in line 81 but not further explained. It is not immediately clear what rollouts are and how they relate to other parts of the method.

Some aspects of the framework remain somewhat unclear to me, for instance:
- Lines 428f. mention that memory items can gradually evolve over time. It is not clear to me how this is possible when no actual consolidation of memory items is happening but when new memory items are simply added. Perhaps the authors mean that memory items evolve into new and somewhat more advanced memory items? The sentence, however, would suggest that REASONINGBANK includes a mechanism that alters the same existing memory items. If so, this should be better explained.
- Line 341 presents “MATTS w/o aggregation” as one of the methods tested in one of the experiments. At this point, it was not entirely clear to me, what the aggregation is and how it works. Perhaps the authors could make this more explicit in some of the previous explanations.

The paper makes some claims that are not entirely credible and could benefit from additional evidence:
- Lines 294ff. make the claim that REASONINGBANK employs a memory extraction strategy that is superior to those of other frameworks. Unlike most other claims in the paper, this claim seems not entirely backed by evidence, as there are no more detailed experiments capable of exactly attributing REASONINGBANK’s outperformance to the memory extraction component. While this appears likely, it could have been other factors such as the memory retrieval, the memory consolidation, or the way that memory items are structured as well.
- The authors present several experiments exploring various aspects of REASONINGBANK and MATTS. The numbers generally look very favourable for their methods, yet some of the experiments are only performed on subsets (e.g., subset of the tested LLMs, subset of the datasets). Not performing all experiments on the full set of all possible combinations is understandable due to budget limitations, however, it is unclear if these restrictions somehow affect the reliability and generalizability of the results. Perhaps, the authors could add at least some error bars or confidence markers to their results to provide an indication of the degree of noise and fluctuation of their results and hence their reliability.

There are also minor issues with the presentation and cosmetics:
- Section 3.1 presents a mathematical problem formalization. I see that the authors likely added this, because some reviewers expect a mathematical formalization in every paper. While the formalization is clear and useful here, it is not used anywhere else in the paper and thus appears as it was simply added pro forma.
- The paper includes a small number of typos, e.g., lines 233 and 238 appear to have been changed and now include incorrect phrases “we generates” and “we refines”.

**Questions:**

Based on the mentioned weaknesses, the following questions can be raised to the authors:
1. How can memory items gradually evolve? What is the underlying mechanism?
2. How does “MATTS w/o aggregation” work? A short explanation may be useful for readers unfamiliar with vanilla TTS.
3. How large are the confidence intervals of the results, especially those results that are only presented for a subset of the LLMs and datasets and how does that affect their reliability and generalizability?

---

> ### Author Response · Authors · 2025-11-18
> **Response to Reviewer WYBc (Part I)**
>
> Thanks so much for your strong, detailed review, and the insightful comments. We feel encouraged that you recognized the originality of our work, with well-crafted implementation details, rigorous and suitable experiments, and the substantial and consistent outperformance our method achieves for a timely and relevant problem in the community. Below, we provide detailed responses to each of your comments and hope to address any further considerations you may have.
>
> ---
>
> > **W1: Issues on the memory consolidation step.**
>
> Thank you for this insightful comment and for the careful reading of our paper. In addition to mentioning our intuition for the current memory consolidation setting in Appendix A.2 (line 927ff), we actually also mentioned this in the **footnote of page 4**.
>
> As you correctly identified, this area is a significant opportunity for future work. We further discuss some potential designs for improving the memory update (consolidation) step here:
> - Hierarchical memory consolidation: The current memory is organized in a linearized way. One promising approach would be tagging each memory with meta data, so that when new experiences are coming, they could be consolidated with respect to corresponding topics/keywords/etc.
> - Memory forgetting/updating: The current "append-only" model is a simplification. A more dynamic "forgetting" or "updating" mechanism would be a key improvement. This could be implemented in several ways. For example, forgetting memory items that are too old in a temporal manner or used less frequently in a useful sense.
>
> ----
>
> > **W2: Precise definition on the two main performance evaluation metrics “success rate” and “average steps” (line 260), specifically, how the success rate is calculated and what constitutes a single step.**
>
> Thanks for your thoughtful review. For these two metrics, we provided a detailed explanation in **Appendix B.1 (line 1017ff) and Appendix B.2 (line 1041ff)**. In the meantime, we agree with your suggestion that a clearer definition on how these two metrics are calculated would help readers better understand the evaluation.
> - Success Rate (SR): A task is marked as ``successful’’ only if the agent’s final output or state precisely matches the pre-defined ground-truth goal, which is measured by the evaluation protocol from the corresponding benchmarks. SR is the total number of successful tasks divided by the total number of tasks evaluated.
> - Average Steps (AS): A single step is defined as one complete agent-env interaction cycle following the ReAct loop, which typically involves observing the current state, generating a thought, and a subsequent action. AS is calculated as the total number of steps taken in the trajectory when solving task.
>
> We have updated the corresponding parts in the revised PDF, please refer to **Appendix B.1 and B.2**.
>
>
> ----
>
> > **W3: The concept of “rollouts” is mentioned in line 81 but not further explained. It is not immediately clear what rollouts are and how they relate to other parts of the method.**
>
> Thank you for pointing out this issue. The standard use of “rollouts” are policy simulations from the current state, used by an agent to evaluate potential actions under uncertainty in order to find the one that maximizes long-term expected value. It was originally developed in reinforcement learning [1]. Later it is quickly adapted to the research line of RLVR [2] and test-time scaling [3], usually referring to sampling multiple trajectories from a given policy. We use the term ``rollout’’ to keep consistency with literature. Following your suggestion, we revised line 80f to make sure it is clearer and better understood for readers from different communities.
>
> [1]  Tesauro, Gerald, and Gregory Galperin. "On-line policy improvement using Monte-Carlo search." Advances in neural information processing systems 9 (1996).
>
> [2]  Guo, Daya, et al. "Deepseek-r1 incentivizes reasoning in llms through reinforcement learning." Nature 645.8081 (2025): 633-638.
>
> [3] Snell, Charlie, et al. "Scaling llm test-time compute optimally can be more effective than scaling model parameters." arXiv preprint arXiv:2408.03314 (2024).

---

> ### Author Response · Authors · 2025-11-18
> **Response to Reviewer WYBc (Part II)**
>
> > **W4 & Q1: How the memory items can gradually evolve over time is not clear (lines 428f), especially when there are no complex memory consolidation steps and memory items are simply added to the ReasoningBank.**
>
> Thanks for your question. This is a qualitative human study. What we intended to describe was not the in-place evolution of a single memory entry, but the evolution of memory knowledge about a specific strategy over time, which manifests as a sequence of new, increasingly sophisticated memories being added to the bank. For example, the strategy of "Strategic Search Query Formulation" has four different yet evolved versions within our ReasoningBank. We have revised **lines 428ff** to make it more precise.
>
> ----
>
> > **W5 & Q2: For “MATTS w/o aggregation” mentioned in line 341, it is not entirely clear what the aggregation is and how it works.**
>
> Thanks for your comment and question. We first want to clarify the concept of “aggregation”. It denotes **whether we are treating the multiple trajectories generated with test-time scaling independently or not**. For MATTS, it induces one set of memory items derived from the aggregation of multiple trajectories. On the other hand, the baseline “MATTS w/o aggregation” induces memory for each trajectory independently. Even if there are multiple trajectories generated, each one creates its own memory based only on its own path, without any information sharing or synthesis from other trajectories, without any “aggregation”. This is equivalent to running ReasoningBank independently multiple times, same as the form of vanilla test-time scaling.
>
> We have revised the corresponding explanation in **Section 3.3** and caption of **Figure 3** to make it clearer.
>
> ----
>
> > **W6: Claims in lines 294ff that “ReasoningBank employs a memory extraction strategy that is superior to those of other frameworks” seems not entirely backed by evidence.**
>
> Thanks for your comment. Our claim in Lines 294ff is indeed supported by evidence. It is based on the fact that in our comparisons against AWM and Synapse, **the memory retrieval and memory consolidation mechanisms were intentionally kept identical to ReasoningBank**. The only significant variable we introduced was our memory extraction strategy (which in turn dictates the memory structure). Therefore, since the other key components were controlled variables, the outperformance we observe can be directly and confidently attributed to our superior extraction method.
>
> We have revised **Appendix B.1** to explicitly state the controlled setting of baseline implementations.
>
> ----
>
> > **W7 & Q3: Reliability and generalizability of the experiment results when some of the experiments are only performed on subsets. May provide an indication of the degree of noise and fluctuation of their results and hence their reliability.**
>
> Thanks for your constructive suggestion. Following your suggestion, we run ReasoningBank and baselines for 5 times on WebArena-Shopping dataset and report the mean and standard deviations in the following table:
>
> | | mean | std |
> | :--- | :--- | :--- |
> | No memory | 38.7 | 1.07 |
> | Synapse | 40.7 | 1.30 |
> | AWM | 44.1 | 1.46 |
> | ReasoningBank | 49.7 | 1.51 |
>
> From the results we can see that ReasoningBank not only achieves a significantly higher mean success rate (49.7) than all baselines, but also demonstrates reliability. ReasoningBank indeed brings higher std than baselines, since there are more stages in our framework. However, the absolute number of std is still low and can confirm that our experiment results are **stable and not subject to significant noise or fluctuation**.
>
> ----
>
> > **W8: Presentation issues with Section 3.1 and minor typos.**
>
> Thanks for your thoughtful comment and for carefully reading our paper.
> For Section 3.1, we appreciate your recognition as “clear and useful”. We humbly believe that this subsection is essential since it introduces the basic formulation of our agent policy and how it interacts with the environment, as well as the trajectory composition. Most importantly, we could clearly define when and where the memory mechanism plays a part. Also, we introduced our testing setting of “test-time learning”, which is another essential aspect. We agree with your feedback that the current presentation does not relate to other parts of the paper, and we have revised Section 3, 4 for callbacks.
>
> For typos in line 233 and 238, we have fixed them accordingly.

---

### Official Review · Reviewer_RD5S · 2025-10-31

**Soundness:** 3
**Presentation:** 4
**Contribution:** 3
**Rating:** 6
**Confidence:** 3

**Summary:**

The authors introduce two key enhancements to existing memory systems. First, they design a memory framework that performs knowledge distillation not only from correct trajectories but also from potentially incorrect ones. Second, during test-time scaling, they propose distilling knowledge from multiple parallel or refinement trajectories (denoted as k trajectories). Their approach is empirically validated through extensive experiments on web navigation and software engineering (SWE) problem-solving benchmarks, demonstrating superior performance over baseline methods. Additionally, ablation studies highlight the contribution of learning from "incorrect" traces.

**Strengths:**

1. The paper presents a memory system that leverages all available logs (both correct and incorrect) and thoughtfully explores its integration with test-time scaling methods.
2. The authors conduct a comprehensive set of experiments and analyses to substantiate their claims.
3. The paper is clearly written and easy to follow.

**Weaknesses:**

1. The proposed memory system may incur high costs since it relies on multiple LLM calls per task trajectory to derive the memory item. Reporting the cost of each method, including memory updates, would help justify the method’s benefits.
2. Because memory updates depend on several LLM calls, the system risks compounding errors—for instance, if the LLM judge incorrectly classifies trajectories or if knowledge distilled from different trajectories fails to generalize.
3. The experiments are limited to the Gemini-2.5 family of models; testing on smaller, publicly available models could strengthen the generalizability of the findings.

**Questions:**

Are there known cases where the LLM judge misclassifies correct versus incorrect traces? If so, what strategies or mechanisms could mitigate such misclassifications?

---

> ### Author Response · Authors · 2025-11-18
> **Response to Reviewer RD5S (Part I)**
>
> Thank you for the positive feedback and encouraging comments. We are glad that you liked our design of the memory system, the comprehensiveness of experiments and analyses, and that our paper is well-written. Below, we provide detailed responses to each of your comments and hope to address any further considerations you may have.
>
> ---
>
> > **W1: The proposed memory system could incur high cost. Reporting the actual cost of each method will help justify ReasoningBank’s benefits.**
>
> Thanks for bringing up the issue of efficiency of agent memory systems. We agree and have already conducted such an evaluation by measuring the number of steps an agent took to solve a task in Tables 1 and 3, in column “averaged steps (AS)”. This shows ReasoningBank usually takes fewer steps and hence is more efficient.
>
> To make the breakdown clearer, we additionally conducted experiments on calculating actual tokens consumed including both input and output by each method in the following table in terms of different stages (averaged token number for each instance in WebArena-Shopping tasks):
>
> | | Action generation | LLM-as-a-Judge | Memory Extraction | Total |
> | :--- | :--- | :--- | :--- | :--- |
> | **No memory** | 50847.4 | - | - | 50847.4 |
> | **Synapse** | 55920.5 | 2594.2 | - | 58514.7 |
> | **AWM** | 53819.6 | 2479.1 | 3074.1 | 59372.8 |
> | **ReasoningBank** | 49306.1 | 2186.3 | 1562.1 | 53054.5 |
>
> From the table we can see that compared with “No memory”, while the total token consumption is **increased only by around 4.3%**, the overall **performance is boosted by 20.5%**. Other memory baselines such as Synapse and AWM will greatly increase the computation overhead while achieving less performance gains compared with ReasoningBank, demonstrating the cost-effectiveness of ReasoningBank.
> We have updated the new results and corresponding analyses in **Appendix C.3**.
>
> ----
>
> > **W2 & Q: Compounding errors introduced in the memory system, e.g., the LLM judge incorrectly classifies trajectories or if knowledge distilled from different trajectories fails to generalize.**
>
> Thanks for pointing out this issue. Following your suggestion, we additionally conduct experiments to study the calibration and robustness on the self-verification step. Following our analysis setting in Section 5, we run experiments with gemini-2.5-flash on the WebArena-Shopping subset.
>
> We first calculated the accuracy for the model’s own self-assessment (LLM-as-a-Judge) using the ground-truth label, which turns out to be 72.7%. To systematically study the robustness/sensitivity of this step, we simulate different accuracies for the self-assessment step from the perfect 100% (ground-truth) down to the worst-case 50% (random-guessing). Detailed descriptions of the setting could be found in **Appendix C. 2 of the revised PDF**. Overall results are presented in the following table for ReasoningBank:
>
> | | 100% (ground-truth) | 90% | 80% | 70% | 60% | 50% (random-gussing) |
> | :--- | :--- | :--- | :--- | :--- | :--- | :--- |
> | **ReasoningBank** | 52.4 | 49.7 | 48.7 | 49.7 | 47.6 | 47.6 |
>
> The results clearly show that the performance of ReasoningBank is highly robust to variations in the self-verification accuracy. The overall accuracy **remains largely stable** within the practical range of $70\%-90\%$ self-judgement accuracy, demonstrating that our approach is not overly sensitive to the quality of the self-judgement signal. This robustness makes ReasoningBank broadly applicable even to tasks where self-verification is challenging or provides non-binary signals.
>
> Additionally, we want to emphasize that the step of explicit self-verification/judgement can be **completely bypassed** using MATTS with the principle of leveraging self-contrastive signals from multiple trajectories while maintaining huge performance improvement.
>
>
> For your question, such cases indeed exist as can be inferred from the accuracy of 72.7%. Here we briefly discuss some potential ways to mitigate this issue and to increase the accuracy of LLM-as-a-Judge. A direct approach would be designing more comprehensive rubrics based on human priors. An advanced version involves collecting trajectory and existing LLM judge results, and training a reward model based on them.

---

> ### Author Response · Authors · 2025-11-18
> **Response to Reviewer RD5S (Part II)**
>
> > **W3: Experiments are limited to Gemini-2.5 families; testing on smaller, publicly available models could strengthen the generalizability of the findings.**
>
> Thanks for your insightful suggestion. To clarify, apart from Gemini-2.5-flash and Gemini-2.5-pro, we also conducted experiments on Claude-3.7-sonnet in the paper and showed that ReasoningBank achieves consistently strong performance.
> We agree with you that testing on smaller, publicly available models would further strengthen our results. Therefore, we conducted experiments using Gemma-3-12B-Instruct (https://huggingface.co/google/gemma-3-12b-it) on the WebArena-Shopping subset. Results are shown in the following table:
>
> | | No memory | Synapse | AWM | ReasoningBank |
> | :--- | :--- | :--- | :--- | :--- |
> | **Success Rate** | 17.1 | 16.0 | 21.4 | 24.1 |
> | **Average Steps** | 13.7 | 14.0 | 12.5 | 11.8 |
>
> We can observe from the results that ReasoningBank still achieves consistent performance gains with much less average interaction steps.

---

### Official Review · Reviewer_u2my · 2025-11-01

**Soundness:** 3
**Presentation:** 3
**Contribution:** 2
**Rating:** 6
**Confidence:** 4

**Summary:**

The paper proposes ReasoningBank, a structured and interpretable memory for continual test-time learning in LLM agents that distills reusable strategies from both successful and failed trajectories. It further introduces memory-aware test-time scaling (MATS), combining parallel and sequential exploration to generate richer, more generalizable memory entries. On WebArena, Mind2Web, and SWE-Bench-Verified, the approach delivers consistent performance and efficiency gains over memory and non-memory baselines, with analyses on memory evolution, the value of failures, and compute efficiency.

**Strengths:**

- **Clear Problem Motivation and Context.** The work identifies that existing memory modules limited learn from failures and disproportionately weight successes, and it clearly argues why this imbalance undermines long-term agent adaptability.

- **Experimental Breadth and Depth.** Evaluation covers diverse agentic tasks (WebArena, Mind2Web, SWE-Bench), several LLM backbones, and robust memory baselines, assessed via effectiveness, step efficiency, and transfer/generalization. Performance comparisons are clear in Tables 1 and 3; Table 4 provides a focused breakdown of success vs. failure step efficiency.

- **Concrete Technical Design.** This method organizes self-verified trajectory analyses into a structured memory of compact, modular strategies, improving both interpretability and reusability.

**Weaknesses:**

- **Ambiguity in Self-Judgment/Verification.** Because trajectory labels rely on the model’s own self-assessment, robustness is uncertain: calibration, error propagation, and blind-spot risks are not quantitatively evaluated. Targeted ablations (e.g., noisy/adversarial self-judgments, third-party or rule-based verifiers) would clarify how sensitive the curation process is.


- **Scalability and Long-Term Lifelong Learning**: Although the paper claims to address "lifelong agents," the experiments are limited to several hundred tasks, and do not simulate truly open-ended deployment or demonstrate the maintenance/pruning of memory over extended horizons. No experiments address issues like memory interference, long-tail drift, or resource constraints on memory expansion.

- **Missing Failure/Robustness/Noise Analysis.** Although Figure 7 illustrates handling of failure trajectories, the work lacks a deeper study of negative cases, when failure-derived memories degrade behavior, and offers limited analysis of noise, redundancy, and their accumulation over time.

**Questions:**

- **Self-Judgment Robustness.** How consistent are the LLM’s success/failure self-labels used for induction? Please analyze misjudged episodes and quantify how such errors propagate to memory quality and downstream performance.

- **Scalability**: Could you provide more detail or share experiments on how ReasoningBank scales as the number of memory items grows—e.g., in lifelong deployment exceeding thousands or millions of tasks?

- **Failure Memory Impact.** Have you observed cases where failure-derived entries bias or degrade behavior (e.g., overfitting to pathological patterns)? What safeguards (filters, calibration, de-duplication) ensure only constructive failures persist?

---

> ### Author Response · Authors · 2025-11-18
> **Response to Reviewer u2my (Part I)**
>
> Thanks for your positive review and constructive comments. We appreciate your recognition of ReasoningBank’s clear motivation and argument, concrete method design, and comprehensive experiments and analyses. Below, we provide detailed responses to each of your comments and hope to address any further considerations you may have.
>
> ---
>
> > **W1 & Q1: Ambiguity and robustness for self-judgement/verification.**
>
> Thanks for your constructive comment. Following your suggestion, we additionally conduct experiments to study the calibration and robustness on the self-verification step. Following our analysis setting in Section 5, we run experiments with gemini-2.5-flash on the WebArena-Shopping subset.
>
> We first calculated the accuracy for the model’s own self-assessment (LLM-as-a-Judge) using the ground-truth label, which turns out to be 72.7%. To systematically study the robustness/sensitivity of this step, we simulate different accuracies for the self-assessment step from the perfect 100% (ground-truth) down to the worst-case 50% (random-guessing). Detailed descriptions of the setting could be found in **Appendix C. 2 of the revised PDF**. Overall results are presented in the following table:
>
> | | 100% (ground-truth) | 90% | 80% | 70% | 60% | 50% (random-gussing) |
> | --- | --- | --- | --- | --- | --- | --- |
> | **ReasoningBank** | 52.4 | 49.7 | 48.7 | 49.7 | 47.6 | 47.6 |
>
> The results clearly show that the performance of ReasoningBank is highly robust to variations in the self-verification accuracy. The overall accuracy **remains largely stable** within the practical range of $70\%-90\%$ self-judgement accuracy, demonstrating that our approach is not overly sensitive to the quality of the self-judgement signal. This robustness makes ReasoningBank broadly applicable even to tasks where self-verification is challenging or provides non-binary signals.
>
> Additionally, we want to emphasize that the step of explicit self-verification/judgement can be **completely bypassed** using MATTS with the principle of leveraging self-contrastive signals from multiple trajectories while maintaining huge performance improvement.
>
> ----
>
> > **W2 & Q2: Scalability of ReasoningBank in long-term life-long learning settings.**
>
> Thanks for your comment. First, we wish to clarify that our paper **does not claim to “address lifelong agents”**. Rather, we only state in our Conclusion (line 485) that ReasoningBank “suggests a practical pathway toward building adaptive and lifelong-learning agents”. Our work primarily serves as a foundational framework that demonstrates the core feasibility and tangible benefit of using reasoning memory for agent self-improvement. We establish that this mechanism works, which is a critical prerequisite before tackling long-term engineering challenges of managing a life-long and extended memory.
>
> When deployed in real-world life-long learning scenarios that incorporate up to millions of tasks, the bottleneck actually lies in memory management. We briefly discuss how we mitigate such challenges here:
> - Our current system relies on embedding-based search, which is a well-studied problem in large-scale information retrieval. For real-world deployment, this retrieval module can be implemented using highly optimized and scalable nearest neighbor search approaches, such as MIPS libraries (e.g., FAISS or ScaNN). These systems are already designed to handle billions of items efficiently, which is well beyond the scale discussed.
> - We can also introduce forgetting or updating operations based on utility frequency and temporal relations. Another promising approach would be tagging each memory item with meta data. In this way, the memory organization would be more hierarchical to pre-filter based on the memory domain (e.g., only retrieve web agent memory for web tasks).

---

> ### Author Response · Authors · 2025-11-18
> **Response to Reviewer u2my (Part II)**
>
> > **W3 & Q3: Missing analyses on potential negative impact of failure-induced memories, degraded behavior, and corresponding safeguards.**
>
> Thanks for your insightful suggestion and for reading our ablation studies in Figure 7. We additionally examined **20 random error cases** that leveraged memory entries induced from failure trajectories.
> Our analysis revealed three primary categories of failure modes introduced by using erroneous memories:
>
> | Category | Frequency (out of 20) | Description |
> | :--- | :--- | :--- |
> | **(i) Incorrect Strategy Extraction** | $\sim 5$ cases | The agent extracts a flawed strategy from a previously misclassified *failed* trajectory (i.e., treating a failure as a success). |
> | **(ii) Misattributed Failure Reasons** | $\sim 6$ cases | The failure reason is misattributed during the extraction process of memory items, leading to sub-optimal strategies. |
> | **(iii) Over-Generalized/Specific Avoidance** | $\sim 9$ cases | Memory items are either too specific or too broad to be leveraged.|
>
> In the majority setting of (iii), we found that the agent actually will not use the memory since the conditions are not met in the over-generalized or over-specific scenario, thus reverting to its baseline (non-memory) behavior.
>
> To mitigate these error cases, we discuss some potential safeguards that could offer insights for future works in this research line. For (i), increasing the accuracy of LLM-as-a-Judge would suffice. To achieve this, one can resort to prompt optimization, detailed rubrics, or even training a reward model. In terms of (ii), an initial solution might include the taxonomy of failure mode, upon which the LLMs could gradually refine the structure and identify the error reason [1].  For (iii), we can further improve memory management by weighting or forgetting each memory item based on utility.
>
> [1] Zhu, Kunlun, et al. "Where LLM Agents Fail and How They can Learn From Failures." arXiv preprint arXiv:2509.25370 (2025).

---

### Author Response · Authors · 2025-11-25
**Thank you for the review and a kind reminder.**

Dear reviewers,

We genuinely appreciate your constructive review and feedback, which greatly helped us to improve our work. We are glad that all reviewers recognized the strengths of ReasoningBank and gave positive initial ratings.

During the rebuttal, we have made great efforts in addressing the concerns through additional experiments, analyses, and clarifications. Based on reviewers’ feedback, we have updated our submission (PDF) **with the differences highlighted in BLUE**. Below is a summary of the key updates:
- Experiments on the calibration and analyses for "LLM-as-a-Judge’" **(Appendix C.2)**.
- Comprehensive inference cost study in addition to average steps **(Appendix C.3)**.
- Case study on emergent capabilities for ReasoningBank **(Appendix C.5)**.
- Clarifications and further explanations on baseline implementation and evaluation metrics **(Appendix B.1)**.
- Refinement on aggregation mechanism for MATTS **(Section 3.3)**.
- Reorganizing and refining findings for original analyses **(Section 5)**.
- Further proofreading and typo fixing.

We sincerely hope that our responses and revisions have addressed the issues you’ve raised satisfactorily and strengthened ReasoningBank as an effective and scalable memory mechanism for LLM agents. We would be happy to discuss with you if there are any further points you wish us to clarify or address for a better score.

Sincerely,

Paper 21820 Authors

---

### Meta-Review · Area_Chair_pYJv · 2026-01-08

**Summary:**

The committee recognizes ReasoningBank as a significant contribution to the field of LLM agents. The paper addresses the critical limitation of agents failing to learn from accumulated experiences by proposing a structured memory framework that distills high-level, reusable reasoning strategies from both successful and failed trajectories. Furthermore, the introduction of Memory-aware Test-Time Scaling (MaTTS) establishes a synergistic loop where memory guides test-time exploration, and diverse explorations, in turn, enrich the memory bank.
The reviewers' primary concerns centered on the robustness of the LLM-as-a-judge mechanism for self-verification, the potential compounding of errors, the computational costs associated with multiple LLM calls during memory extraction, and the generalizability of the findings beyond the Gemini model family. The decision to accept is informed by the authors' thorough rebuttal, which provided empirical evidence of the system's robustness to verification noise and its cost-effectiveness compared to existing memory baselines.

**Reviewer Concerns:**

Several reviewers (u2my, RD5S, sUDi) questioned the reliability of using an LLM to judge its own success/failure. The authors successfully addressed this by conducting a calibration study (Appendix C.2), demonstrating that ReasoningBank's performance remains stable even when the self-judgment accuracy drops significantly, proving the system's resilience to verification noise.

**Reviewer Scores:**

Based on the post-rebuttal discussion and the depth of the revised manuscript, I think all reviewers will maintain a positive evaluation.

---

### Decision · Program_Chairs · 2026-01-26

Accept (Poster)